# Imaging of brain electric field networks with spatially resolved EEG

**Lawrence R Frank**[1*†, ‡], **Vitaly L Galinsky**[1†], **Olave Krigolson**[2§], **Susan Tapert**[1#], **Stephan Bickel**[3,4¶, **], **Antigona Martinez**[4**]

[1]University of California, San Diego, La Jolla, United States; [2]University of Victoria, Victoria, Canada; [3]Nathan Kline Institute, Orangeburg, United States; [4]Feinstein Institute for Medical Research, New York, United States

*For correspondence:
lfrank@ucsd.edu

Present address: [†]Department of Radiology, University of California San Diego, La Jolla, United States; [‡]Centerfor Functional MRI, Department of Radiology, University of California San Diego, La Jolla, United States; [§]Centre for Biomedical Research, University of Victoria, Victoria, Canada; [#]Dept of Psychiatry, UC San Diego, La Jolla, United States; [¶]Nathan Kline Institute, Orangeburg, United States; [**]The Feinstein Institutes for Medical Research, Northwell Health, Manhasset, United States

Competing interest: The authors declare that no competing interests exist.

## eLife Assessment

This **fundamental** work has the potential to advance our understanding of brain activity using electrophysiological data, by proposing a completely new approach to reconstructing EEG data that challenges the assumptions typically made in the solutions to Maxwell's equations. **Convincing** evidence for the superior spatio-temporal resolution of this method is provided through a number of experiments, including simultaneous FMRI/EEG acquisitions. This work will be of broad interest to neuroscientists and neuroimaging.

**Abstract** We present a method for spatially resolving the electric field potential throughout the entire volume of the human brain from electroencephalography (EEG) data. The method is *not* a variation of the well-known 'source reconstruction' methods, but rather a direct solution to the EEG inverse problem based on our recently developed model for brain waves that demonstrates the inadequacy of the standard 'quasi-static approximation' that has fostered the belief that such a reconstruction is not physically possible. The method retains the high temporal/frequency resolution of EEG, yet has spatial resolution comparable to (or better than) functional MRI (fMRI), without its significant inherent limitations. The method is validated using simultaneous EEG/fMRI data in healthy subjects, intracranial EEG data in epilepsy patients, comparison with numerical simulations, and a direct comparison with standard state-of-the-art EEG analysis in a well-established attention paradigm. The method is then demonstrated on a very large cohort of subjects performing a standard gambling task designed to activate the brain's 'reward circuit'. The technique uses the output from standard extant EEG systems and thus has potential for immediate benefit to a broad range of important basic scientific and clinical questions concerning brain electrical activity. By offering an inexpensive and portable alternative to fMRI, it provides a realistic methodology to efficiently promote the democratization of medicine.

## Introduction

The human brain communicates internally through exceedingly complex spatial and temporal patterns of electrical signals. Although these signals can be measured using electrodes placed on the surface of the scalp (electroencephalography [EEG]), the ability to reconstruct the spatial and temporal patterns within the brain has been thwarted by the complexity of the inverse problem: What time- (or frequency-) dependent volumetric electrical signals throughout the brain are consistent with the signal measured on the two-dimensional surface of the scalp (*Marinazzo et al., 2019*; *Michel and Brunet, 2019*)? There is a long-standing belief that it is not possible to detect and reconstruct electrical activity in subcortical regions deep within the brain from EEG due to inherent limitations

of 'volume conduction' (*Nunez et al., 1997*). However, this is not actually a physical limitation, but rather a consequence of the incomplete nature of the standard model used to characterize the EEG signal. Despite the obviously highly dynamical nature of the electrical activity that occurs within the very inhomogeneous and anisotropic composition of brain tissue, current EEG data analysis methods are still based on the assumption that the average tissue bioelectric properties (e.g. the average permittivity $\bar{\epsilon}_0$ and conductivity $\bar{\sigma}$) are sufficient to describe the electric fields $E$ in the brain. This leads to the approximation $|\bar{\epsilon}_0 \partial E/\partial t| \ll |\bar{\sigma} E|$ (*Taulu and Larson, 2021*), which in turn leads to the assumption that the time dependence $\partial E/\partial t$ can be ignored in the 'typical' frequency range of brain signals (*Hämäläinen et al., 1993*). This is the ubiquitous so-called 'quasi-static' approximation (*Gaugain et al., 2023*; *Rapetti and Rousseaux, 2014*).

In reality, it is precisely the anisotropic and inhomogeneous nature of brain tissue that must be taken into account in order to develop an accurate physical model of brain electromagnetic (EM) behavior, as we have described in our recently developed universal theory of brain waves called *weakly evanescent transverse cortical waves* (WETCOW) (*Galinsky and Frank, 2020b*; *Galinsky and Frank, 2021*). The surprising consequence of this theory is the existence of electric field waves generated as a consequence of the complex tissue boundaries (e.g. surface waves) that permeate throughout the brain and are in precisely the frequency range of observed brain electrical activity. This theory explains the broad range of observed but seemingly disparate brain spatiotemporal electrical phenomena from extracellular spiking to cortical wave loops, all of which are predicated on the time dependence of the electric fields within the complex architecture of anisotropic and inhomogeneous tissue within the brain. This theory is necessary to provide a solution to the EEG inverse problem which, as shown below, produces a reconstruction of brain electrical activity with high temporal resolution and spatial resolution that is comparable to (or even exceeding that of) functional MRI (fMRI).

## Results

## A new physical theory of brain waves

### Background

The fact that the brain produces electrical signals, or brain waves, has been known for over 150 years, and the first recording in humans using EEG was made almost 100 years ago. The pioneering work of Cajal in the late 19th century established the neuron doctrine that the nervous system is made up of discrete individual cells (neurons), which is one of the central tenets of modern neuroscience. Neurons are known to generate electrical signals as a result of their ability to maintain a voltage difference across their membranes that generates an electrochemical pulse known as an action potential that can travel rapidly along the axon. Consequently, the majority of approaches to characterizing brain dynamical behavior are based on the assumption that signal propagation along well-known anatomically defined pathways, such as major neural fiber bundles, tracts, or groups of axons, should be sufficient to deduce the dynamical characteristics of brain activity at different spatiotemporal scales. Characterizing brain networks is important for understanding many aspects of brain function, from neural processes underlying cognition to aberrant brain electrical activity, such as seen in epileptic seizures.

However, this view cannot explain the entire picture of observed brain activity propagation. Recently spatiotemporally organized, circular wave-like patterns of electrophysiological activity (*traveling waves*) were described at the macroscopic (scalp EEG, MEEG) and mesoscopic scale (invasive EEG), in animal models and humans, and during cognitive tasks and sleep (*Muller et al., 2018*; *Zhang et al., 2018*; *Zanos et al., 2015*). These findings represent a formidable challenge for current network theories to explain such a remarkable synchronization across a multitude of different local networks.

In the following sections, we introduce the basic physical problem, present the rationale behind the ubiquitous 'quasi-static approximations', and detail why it is a poor model for brain activity, and then outline our recently developed more general universal theory of brain waves and how it ultimately leads to a solution of the inverse problem for EEG data.

### Maxwell's equations in the brain

The general equations governing the propagation of EM waves are called Maxwell's equations (*Maxwell, 1865*) and in an inhomogeneous and anisotropic medium take the form

$$\nabla \cdot \boldsymbol{D} = \rho \tag{1a}$$

$$\nabla \cdot \boldsymbol{H} = 0 \tag{1b}$$

$$\nabla \times \boldsymbol{E} = -\frac{\partial \boldsymbol{H}}{\partial t} \tag{1c}$$

$$\nabla \times \boldsymbol{H} = \boldsymbol{J}_t \tag{1d}$$

where $\boldsymbol{D} = \varepsilon \boldsymbol{E}$ is the *electric displacement field*, where the (scalar) *permittivity* $\varepsilon$ takes into account the polarization of the dielectric material in the electric field $\boldsymbol{E}$, $\boldsymbol{H}$ is the magnetic field intensity, and the *total current density* $\boldsymbol{J}_t$ is given by the sum of the free current density $\boldsymbol{J}_f$ and bound current density $\boldsymbol{J}_b$:

$$\boldsymbol{J}_{total} = \boldsymbol{J}_{free} + \boldsymbol{J}_{bound} = \boldsymbol{J}_{Conductive} + \boldsymbol{J}_{Displacement} \tag{2}$$

where

$$\boldsymbol{J}_C = \boldsymbol{\sigma} \cdot \boldsymbol{E} \quad , \quad \boldsymbol{J}_D = \frac{\partial \boldsymbol{D}}{\partial t} \tag{3}$$

are the *conductive current* and *displacement current*, respectively.

This is the problem setup. To understand the electric fields in the brain, one needs to solve Maxwell's equations. At this point, the standard procedure (e.g. **Nunez et al., 1997**) is to simply ignore the temporal variations in both the magnetic field and the electric field by setting $\frac{\partial \boldsymbol{H}}{\partial t} = 0$ and $\frac{\partial \boldsymbol{E}}{\partial t} = 0$. This simplified **Equation 1c** to $\nabla \times \boldsymbol{E} = 0$ and **Equation 1d** to $\nabla \times \boldsymbol{H} = \boldsymbol{J}_C$ since eliminating the time dependence of the electric field eliminates the displacement current: $\boldsymbol{J}_D = \frac{\partial \boldsymbol{D}}{\partial t} = \varepsilon \frac{\partial \boldsymbol{E}}{\partial t} = 0$. This is the so-called *quasi-static approximation* (**Gaugain et al., 2023**; **Rapetti and Rousseaux, 2014**) ubiquitous in EEG analysis methods.

What are the justifications for these simplifications? It turns out that in biological tissues, the inductive effects are small or negligible (**Plonsey and Heppner, 1967**), so that eliminating the time dependence of the magnetic field is indeed justified. This is important as the simplified form of **Equation 1c** implies, for simple vector relations, that the electric field can be written in terms of a *field potential* $\phi$ since

$$\nabla \times \boldsymbol{E} = 0 \Rightarrow \boldsymbol{E} = -\nabla \phi \tag{4}$$

Solving for the electric field $E$ is then equivalent to solving for $\phi$. And we can ignore any magnetic field effects.

However, the assumption that the electric field does not vary with time is *not* justified in biological materials (**Plonsey and Heppner, 1967**), and, therefore, the expression in **Equation 1d** is correct as is - the displacement current must be retained. Maxwell's equations in the brain thus take on a somewhat odd configuration in that they are, in the standard physics parlance, *magnetostatic* but not *electrostatic*.

It is somewhat ironic that the introduction of the displacement current, which was in some sense Maxwell's greatest insight and the final piece of the puzzle in solving the equations of electromagnetism, turns out to be the key to the puzzle of brain activity, where it had once again been ignored.

## Consequences of the quasi-static approximation

Because of the ubiquity of the quasi-static approximation, it is worth pausing here to consider its consequences, since in our view they have led to confusion in the understanding of brain electrical activity and the problem of EEG reconstruction.

$$\left. \begin{array}{l} E = -\nabla \phi \\ \nabla \cdot \boldsymbol{E} = \rho/\varepsilon \end{array} \right\} \quad \Rightarrow \quad \nabla^2 \phi = \frac{\rho}{\varepsilon}$$

which is Poisson's equation and relates the electric field potential $\phi$ to 'sources' $\rho$. The most striking aspect of this solution, though it was the obvious endpoint by construction, is that there is no time dependence in the solution. One would have guessed this to be a giant red flag for the description of brain electrical activity, but the persistence of this approach has nevertheless been tenacious. Consequently, from the perspective of EEG reconstruction, the problem is framed in terms of 'source

reconstruction'. We note that these equations are also called the 'quasi-static volume conduction' equations, and, therefore, this problem is often referred to as the 'volume conduction' problem.

There is also another massive source of confusion that is often used to justify the quasi-static approximation. The logic goes something like this. We know that the brain has, for example, alpha waves which for the sake of simplicity we will assume the frequency to be a typical value of $\omega = 10Hz$. For EM waves in a medium of permittivity $\varepsilon$, the wavelength of these waves is related to the velocity $v$ of the waves as

$$\lambda = \frac{v}{\omega} \qquad \text{where} \qquad v = \frac{c}{\sqrt{\varepsilon}} \tag{5}$$

in which $c = 3 \times 10^8 \, m/s$ is the speed of light. For a typical tissue permittivity $\varepsilon = 100$, the wave velocity is $v = 3 \times 10^7 \, m/s$ so that the wavelength is

$$\lambda = \frac{v}{\omega} = \frac{3 \times 10^7}{10} = 3 \times 10^6 m = 3000 \, km \tag{6}$$

Because the wavelength is so much greater than the spatial dimensions of the head, there can be no appreciable phase difference anywhere in the head, and EM wave propagation effects can be ignored. Indeed, this is true - there are effectively no EM wave propagation effects in the brain. But that argument is *not* a justification for the assumption of a time-independent electric field. Indeed, the logic is backward. One must first solve Maxwell's equations under the proper conditions, then eliminate contributions that appear insignificant.

Indeed, if one simply assumes the absence of free charges, Maxwell's equations *Equation 1c* and *Equation 1d* in a medium of permittivity $\varepsilon$ and permeability $\mu$ combine to give

$$\nabla^2 \boldsymbol{E} - \frac{1}{v^2} \frac{\partial^2 \boldsymbol{E}}{\partial^2 t} = 0 \quad \Rightarrow \quad \boldsymbol{E}(\boldsymbol{r}, t) = \boldsymbol{E}_0 e^{i(\boldsymbol{k} \cdot \boldsymbol{r} - \omega t)} \tag{7}$$

the solution to which are complex plane waves, which in turn implies the *dispersion relation* $\omega = v \, |\boldsymbol{k}|$. As we will show below, the dispersion relation derived from the correct version of Maxwell's equations is quite different and provides a key insight into the interesting characteristics of brain waves.

The fact that observed alpha waves have velocities many orders of magnitude slower than EM alpha waves should be an obvious clue that something else is going on. This much is recognized in that they are ascribed to ill-defined concepts such as 'neuronal oscillations'. But a consequence of that should be a reexamination of Maxwell's equations in light of these experimental observations. We will do that in the next section and demonstrate that these 'slow' waves are not mysterious at all, but a direct consequence of the displacement current and the inhomogeneity and anisotropy of the tissues. They are not EM waves, but surface waves.

## The general solution: WETCOW theory

In this section, we provide a brief outline of the more detailed theoretical description in *Galinsky and Frank, 2020a*; *Galinsky and Frank, 2020b*. From the above discussion, the proper form of Maxwell's equations to solve, from *Equation 1a and d* and *Equation 3*, is

$$\left. \begin{array}{l} \nabla \cdot \boldsymbol{D} \quad = \rho \\ \nabla \times \boldsymbol{H} \quad = \boldsymbol{\sigma} \cdot \boldsymbol{J}_c + \dfrac{\partial \boldsymbol{D}}{\partial t} \end{array} \right\} \quad \Rightarrow \quad \frac{\partial \rho}{\partial t} + \nabla \cdot \boldsymbol{J}_c$$

where the RHS is a statement of *charge continuity*. These equations along with *Equation 4* and *Equation 3* give the charge continuity equation in terms of the quantity of interest, the electric field potential $\phi$:

$$\frac{\partial}{\partial t} \left( \nabla^2 \phi \right) = -\nabla \cdot \boldsymbol{\Sigma} \cdot \nabla \phi \tag{8}$$

where $\boldsymbol{\Sigma} = \{\sigma_{ij}/\varepsilon\}$ is the scaled conductivity tensor. Thus, the inclusion of the displacement current has produced a wave equation.

A simple linear wave analysis, i.e., substitution of $\phi \sim \exp\left[-i(\boldsymbol{k} \cdot \boldsymbol{r} - \Omega t)\right]$, where $\boldsymbol{k}$ is the wave number, $\boldsymbol{r}$ is the coordinate, $\Omega$ is the frequency, and $t$ is the time, gives the following complex dispersion relation, now written in tensor form where $i, j = \{x, y, z\}$ and repeated indices are summed:

$$D(\Omega, \boldsymbol{k}) = -i\boldsymbol{\Omega}k_i^2 - \Sigma_{ij}k_ik_j - i\partial_i\Sigma_{ij}k_j = 0, \tag{9}$$

which is composed of the real and imaginary components:

$$\gamma \equiv \mathfrak{I}[\Omega] = \Sigma_{ij}\frac{k_ik_j}{k^2} \qquad \omega \equiv \mathfrak{R}[\Omega] = -\frac{\partial_i\Sigma_{ij}k_j}{k^2} \tag{10}$$

Several interesting features of this relation are worth noting. Because it is complex, it will result in both an oscillatory component (proportional to the frequency $\omega$) and a decaying component (proportional to the decay rate $\gamma$). Both $\gamma$ and $\omega$ are functions of the tissue parameters through $\Sigma_{ij}$ so there is a direct connection between tissue properties and the wave dynamics. The tissue properties are encoded in the tensor $\Sigma_{ij}$ so spatial variations due either to inhomogeneity or anisotropy will also influence wave propagation. But perhaps the most interesting feature of this dispersion relation is that $\omega \sim 1/k$, and so quite different than the dispersion relation for EM waves. This has significant consequences for the nature of brain electrodynamics, as shown below.

These results bring us to a central important point. For typical low-frequency ($\lesssim 10Hz$) *average* values of white (WM) and gray matter (GM) conductivity and permittivity (i.e. from *Gabriel et al., 1996a*; *Gabriel et al., 1996b*), the decay rates give strong wave damping, and no waves would be observed. For example, typical values for GM and WM are $\varepsilon_{GM} = 4.07 \cdot 10^7 \varepsilon_0$, $\varepsilon_{WM} = 2.76 \cdot 10^7 \varepsilon_0$, $\sigma_{GM} = 2.75 \cdot 10^{-2}$ S/m, $\sigma_{WM} = 2.77 \cdot 10^{-2}$ S/m, where $\varepsilon_0 = 8.854187817 \cdot 10^{-12}$ F/m is the vacuum permittivity so the damping rate $\gamma$ is in the range of 75–115 s$^{-1}$, which would give strong wave damping. This leads immediately to the question of the effects of the anisotropy, which is encoded in the scaled conductivity tensor $\Sigma$. A full discussion of the effects of anisotropy is provided in *Galinsky and Frank, 2020a*; *Galinsky and Frank, 2020b*, but, here, we review the key novel and important finding of the general theory: the existence of previously unrecognized (at least theoretically) waves *transverse* to the fiber direction.

To see this, we take a very simple idealized tissue model: fibers are packed in a half space aligned in $z$ direction, and their number decreases in $x$ direction in a relatively thin layer at the boundary. We assume that small cross fiber currents can be characterized by a small parameter $\epsilon$ and represent the conductivity tensor as

$$\boldsymbol{\Sigma} = \begin{pmatrix} \epsilon v & \epsilon v & \epsilon v \\ \epsilon v & \epsilon v & 0 \\ \epsilon v & 0 & v \end{pmatrix}. \tag{11}$$

where $v \equiv v(x)$. For the $v(x)$ dependence, we will assume that the conductivity is changing only through a relatively narrow layer at the boundary, and the conductivity gradient is directed along $x$ axis.

$$\left([\partial_t + a]\partial_z^2 + b\partial_z\right)\phi_{\parallel} = 0 \qquad , \qquad \sim \epsilon^0 \quad \text{(damped oscillator)} \tag{12a}$$

$$\left(\partial_t\partial_y^2 + b\partial_y\right)\phi_{\perp} = 0 \qquad , \qquad \sim \epsilon^1 \quad \text{(wave equation)} \tag{12b}$$

where $a = v(x)|_{x_0}$ and $b = \partial_x v(x)|_{x_0}$ are considered constant evaluated at the boundary $x_0$, and $\epsilon^0$ and $\epsilon^1$ denote the zeroth and the first orders of $\epsilon$ power. We emphasize that this approximation for $a$ and $b$ is specifically allowed because we are considering a thin boundary layer problem.

The first equation, *Equation 12a*, describes a potential along the fiber direction and is a damped oscillator equation that has a decaying solution. But the second equation, *Equation 12b*, describes a potential perpendicular to the fiber direction and does not include a damping term; hence, it describes a pure wave-like solution that propagates in the thin layer transverse to the main fiber direction. Thus, although this wave-like solution $\phi_{\perp}$ has a smaller amplitude than along the fiber action potential $\phi_{\parallel}$,

it can nevertheless have a much longer lifetime. Such waves are called *weakly evanescent transverse cortical waves*, or WETCOW for short.

Though these produce many interesting effects, two aspects are most important for the current application. First, the decay rates $\gamma$ for brain tissue are sufficiently small that waves can persist for time significantly longer than 'spiking'. The persistence of the stable waves can be characterized by the ratio of the decay rate to the frequency, which from simple geometric considerations from *Equation 10* for the longest waves (with the smallest amount of damping) with

$$\frac{\gamma}{\omega} = \frac{\Sigma_{ij}k_ik_j}{\partial_i\Sigma_{ij}k_j} \approx\sim 0.02 - 0.04.$$

Anisotropy ($\Sigma_\perp < \Sigma_\parallel$) will reduce this estimate even further (see *Galinsky and Frank, 2020a*; *Galinsky and Frank, 2020b*, for more details). In other words, anisotropy can result in decay rates that can vary from these maximum (homogeneous) values above, all the way down to 0, based on the direction of propagation, increasingly supporting the existence of transverse waves. Without taking anisotropy into account, i.e., assuming the mean tissue values above as is done in the 'standard model', the decay is so rapid that transverse weakly evanescent waves are not supported.

Second, the inverse relationship between the frequency and wavelength in the dispersion relation (*Equation 9*) means that waves can extend throughout the entire volume of the brain. One can also recognize immediately from *Equation 10* the existence of significant phase variations across the brain (characterized by both phase and group velocity) proportional to tensor products $\nabla \cdot \boldsymbol{\Sigma}$ and $\nabla \cdot \boldsymbol{\Sigma} \cdot \boldsymbol{k}$ that characterize wave propagation normal to the conductivity gradient and thus normal to the fiber orientation. This contradicts the long-standing belief that there are no significant phase variations across the head. There are, but they are not due to EM waves, but WETCOW waves.

The existence of these waves has profound implications for the understanding of brain electrical activity and communications and has been shown to explain a wide range of observed collective brain behaviors, including spiking in the extracellular space (*Galinsky and Frank, 2020a*; *Galinsky and Frank, 2020b*), rapid signal synchronization (*Galinsky and Frank, 2021*) that provides a mechanism for learning and memory (*Galinsky and Frank, 2023b*), and neuronal avalanches (*Galinsky and Frank, 2023a*; *Galinsky and Frank, 2023c*). And of course, they require rethinking what is meant by a brain 'network', since signal propagation must now be considered not only along fiber pathways, but between structures that may not even be neuronally directionally connected. But for the present purposes, they imply the existence of waves of electrical activity throughout the brain.

## Solution to the inverse EEG problem

### Theory

The WETCOW theory predicts the existence of waves satisfying Maxwell's equations in the brain where the morphology and tissue characteristics have been properly taken into account. The EEG inverse problem therefore involves estimating the electric field potential $\phi$ that satisfies Maxwell's equations constructed with the tissue properties of a particular brain, satisfying boundary conditions determined by the morphology of the brain, and consistent with measurements made in an array of electrodes on the surface of the brain. Our method for solving the inverse EEG problem can be summarized as follows. Given a standard EEG dataset from $N$ electrodes and a high-resolution anatomical (HRA) MRI dataset with high contrast between GM and WM, the solution to the inverse EEG problem can be formulated as an approximation for the volumetric distribution of electrostatic potential inside the complex inhomogeneous and anisotropic tissues and complicated morphology of the MRI domain (*Galinsky et al., 2018*).

The solution to the EEG inverse problem entails solving *Equation 8* for the electric field potential $\phi$. Taking the temporal Fourier transform (i.e. replacing $\partial/\partial t \rightarrow -I\omega$, $I^2 = -1$, where $\omega$ is the frequency), the electrostatic potential satisfies the equation in the Fourier (i.e. frequency) domain, and using the notation $\partial_i = \partial/\partial q_i$, $q_i = \{x, y, z\}$, can be written in tensor form as

$$\left(\Sigma^{ij} - I\omega\varepsilon\delta^{ij}\right)\partial_i\partial_j\phi_\omega = \left[I\omega(\partial_i\varepsilon)\delta^{ij} - (\partial_i\Sigma^{ij})\right]\partial_j\phi_\omega + \mathcal{F}_\omega, \tag{13}$$

where $\delta$ is the Dirac delta function, and a summation is assumed over repeated indices. This can be expressed in the form $\hat{L}\phi_\omega = \hat{R}\phi_\omega + \hat{\mathcal{F}}_\omega$ in terms of the operators $\hat{L} \equiv \partial^i \partial_i$, a frequency-dependent source term $\hat{\mathcal{F}}_\omega$ and the operator

$$\hat{R} \equiv \frac{\sigma + I\omega\varepsilon}{\sigma^2 + \omega^2\varepsilon^2} \left[ I\omega(\partial_i\varepsilon)\delta^{ij} - (\partial_i\Sigma^{ij}) - \left(\Sigma^{ij} - \sigma\delta^{ij}\right)\partial_i \right] \partial_j$$

where $\mathbf{\Sigma} = \{\Sigma^{ij}\}$ is a local tissue conductivity tensor and $\sigma = Tr\mathbf{\Sigma}/3 = \Sigma_i^i/3$ is an isotropic local conductivity. Terms in square brackets show that the parts of $\hat{R}\phi_\omega$ can be interpreted in terms of different tissue characteristics and may be important for understanding the origin of sources of the electro-/magnetostatic signal detected by the EEG sensors. The first term ($\omega(\partial^i\varepsilon)(\partial_i\phi_\omega)$) corresponds to areas with sudden change in permittivity, e.g., the WM/GM interface. The second term ($(\partial_i\Sigma^{ij})(\partial_j\phi_\omega)$) corresponds to regions where the conductivity gradient is the strongest, i.e., the GM/CSF (cerebral spinal fluid) boundary. Finally, the last term ($\Sigma^{ij}\partial_i\partial_j\phi_\omega - \sigma\partial^i\partial_i\phi_\omega$) includes areas with the strongest conductivity anisotropies, e.g., input from major WM tracts. The frequency- and position-dependent internal sources $\hat{\mathcal{F}}_\omega$ can be used to incorporate various nonlinear processes, including multiple frequency effects of the efficient synchronization/desynchronization by brain waves or effects of their critical dynamics. This term is ignored in the current paper because they are higher-order terms that complicate the processing (and interpretation) but do not substantially change the main results. They will be considered in future work.

## Numerical implementation

The inverse problem can be solved by constructing an approximate solution for the potential $\phi$ across an entire brain volume iteratively as $\hat{L}\phi_\omega^{(k)} = \hat{R}\phi_\omega^{(k-1)}$ and $\tilde{\phi}_\omega^{(K)} = \alpha_K \sum_{k=0}^{K} \phi_\omega^{(k)}$ (**Galinsky et al., 2018**), where a single iteration forward solution is found using a Fourier-space pseudo-spectral approach (**Gottlieb and Orszag, 1977**). The volumetric frequency-dependent potential $\tilde{\phi}_\omega^{(K)}$ is the central quantity of interest, and it can be calculated over arbitrary frequency ranges $\omega = \omega_1 \ldots \omega_2$, such as the standard frequency bands of interest in EEG. These potentials can then be converted to the time domain $\tilde{\phi}(t, \boldsymbol{x})$ from which space-time modes can be determined using our *entropy field decomposition* (EFD) method for analysis for complex nonlinear systems (**Frank and Galinsky, 2016a**; **Frank and Galinsky, 2016b**) (see Appendix 1: The entropy field decomposition). Alternatively, as in this work, the estimated potentials $\tilde{\phi}(t, \boldsymbol{x})$ can be used in the joint estimation scheme presented in **Galinsky and Frank, 2019**, as an additional modality $Q_{ij}^E$ in the intermodality coupling matrix $\mathcal{Q}_{ij}$ (see Appendix 1: Multi-modality EFD (JESTER)). The potential depends upon the electrical properties of the tissue permittivity, permeability, and conductivity. These parameters can be estimated from the HRA MRI data. Using *joint estimation with entropy regularization* (JESTER), data from MRI can be used to define the complex brain tissue morphology and constrain the tissue-specific values of $\mathbf{\Sigma}$ and $\varepsilon$. This procedure of inverting the WETCOW brain wave model constrained by MRI-defined tissue properties is called *SPatially resolved EEG Constrained with Tissue properties by Regularized Entropy* (SPECTRE).

An approximate pseudo-spectral solution for the potential $\phi$ was constructed across an entire brain volume using either MRI Montreal Neurological Institute (MNI) 2 mm resolution ($91 \times 109 \times 91$ voxel dimensions), 1 mm MNI resolution ($182 \times 218 \times 182$ voxel dimensions), or 0.7 mm resolution ($207 \times 256 \times 215$ voxel dimensions). For the current study, we only used the anatomical data for estimation and assignment of different tissue types, and no diffusion MRI (dMRI) data was used. To register between different modalities, including MNI, HRA, fMRI, etc., and to transform the tissue assignment into an appropriate space, we used the *symplectomorphic registration* (SYMREG) registration method (**Galinsky and Frank, 2019**).

The pseudo-spectral computational approach used in SPECTRE has some important advantages over the finite/boundary element approaches typically used for electrostatic modeling of brain activity (**Gramfort et al., 2010**; **Kybic et al., 2005**; **von Ellenrieder et al., 2009**; **Gutiérrez and Nehorai, 2008**; **Schimpf et al., 2002**; **Ermer et al., 2001**; **Mosher et al., 1999**). It does not use surface meshes and so does not require limiting the location of activity sites to a small number of surfaces with fixed number of static dipole sources constrained to the surfaces. And the distribution of both electrostatic and geometric properties of the media (conductivity, permittivity, anisotropy, inhomogeneity - derived from the MRI data) is incorporated at every location throughout the volume. It is thus able to find a time-dependent spatial distribution of the electrostatic potential at every

space-time location of a multidimensional volume as a superposition of source inputs from every voxel of the same volume (*Galinsky et al., 2018*). These traits allow it to model wave-like signal propagation inside the volume and can detect and characterize significantly more complex dynamical behavior of the sources of the electrostatic activity recorded at the sensor locations than traditional methods.

## Methods

### Summary of SPECTRE

The SPECTRE procedure can be summarized as follows. The data are the raw output from a standard EEG system and an HRA MRI image. A standard template (e.g. T1-weighted anatomical MNI [*Fonov et al., 2009*]) is typically used so that an MRI acquisition is not required. The EEG data is registered to the HRA template using our nonlinear SYMREG (*Galinsky and Frank, 2019*). The different tissue types and their geometry are determined from the HRA using our *spherical wave decomposition* algorithm (*Galinsky and Frank, 2014*). The estimated geometry is used to define the sampling points for the pseudo-spectral algorithm. The spatial variations in the tissue bioelectric properties are estimated from the spatial variation in the segmented tissue types. The pseudo-spectral algorithm is then solved for the electric field potential that best fits the raw EEG data at each electrode, constrained by the local tissue properties within the brain volume. The resulting potential field $\phi(\boldsymbol{x}, t)$ is then decomposed into spatial-temporal modes using the EFD algorithm (*Frank and Galinsky, 2016a*; *Frank and Galinsky, 2016b*) constrained by the anatomical atlas using JESTER (*Galinsky and Frank, 2017*).

In the present study, the HRA data were used to identify and segment the GM and WM regions in order to define their separate geometries and the spatial variations in the tissue bioelectric properties (e.g. conductivity and permittivity) to go into the estimation of the field potential. However, SPECTRE is quite flexible in its ability to incorporate additional tissue information from other modalities, so as to improve estimates of the local tissue conductivity tensor from dMRI data, where it is available. We did not do so in the current study as the goal was to demonstrate that SPECTRE can be achieved without the necessity of acquiring any MRI data, which has significant practical implications.

The conductivity tensor is not exactly the same as the diffusion tensor in brain tissues, but they are closely related. While both tensors describe transport properties in brain tissue, they represent different physical processes. The conductivity tensor is often assumed to share the same eigenvectors as the diffusion tensor. There is a strong linear relationship between the conductivity and diffusion tensor eigenvalues, as supported by theoretical models and experimental measurements. For the current study, we only used the anatomical data for estimation and assignment of different tissue types, and no dMRI data was used.

To understand intuitively why SPECTRE is capable of reconstructing EM activity through the entire brain, including deep within subcortical structures, a simple idealized example is helpful. Consider two point current sources of different frequencies, one in the cortical layer close to the scalp, the second deep within the subcortical structures of the brain. Consider a single sensor placed on the scalp collinear with the two sources. Standard source localization methods will not see the deep source, since there is no frequency dependence, and the signal falloff is simply a function of the distance from the sensor. Therefore, the close source completely dominates the signal model. Since all tomographic imaging methods (e.g. MRI, CT, etc.) depend strongly on both the spatial and temporal sampling of the measured physical system, this effective invisibility of currents in the standard quasi-static model essentially precludes the solution of the true inverse EEG problem and necessitates the artificial construction of assumed dipole distribution on pre-chosen artificial internal structures. In contrast, in SPECTRE, the sources are not dipoles, but frequency sources that extend throughout the entire brain volume subject to the boundary conditions imposed by both the tissues geometry and its spatially and frequency-dependent properties. The surface electrodes are assumed to be sensing EM waves emanating from the entire brain across a broad-frequency spectrum limited only by the sensors. Used in conjunction with an HRA MRI data that provides the spatial distribution of the frequency-dependent tissue electrical properties that constrain the possible solution, SPECTRE can invert the wave equations to provide an estimate of the spatiotemporal distribution of the electric field potential.

## Mode reconstruction

After estimating the nonlinear spatially and temporally varying electric field potential, we still face the challenge of interpreting it, much like raw fMRI data must be analyzed to identify activation patterns. At this stage, the issues of fMRI and SPECTRE analysis are essentially the same. In general, this is a difficult task because brain activity exhibits a highly complex spatiotemporal structure. Conceptually, one can view 'activation patterns' (or modes) as groups of spatially contiguous voxels sharing similar time courses, which may synchronize with other local regions located anywhere else in the brain. For example, the 'default mode' network comprises several such contiguous regions - such as the dorsal medial prefrontal cortex (PFC), posterior cingulate cortex, precuneus, and angular gyrus - that operate together.

It is important to highlight several complicating factors inherent in time-dependent volumetric data from modern imaging systems, including neuroimaging scanners and meteorological radar. First, estimating spatiotemporal patterns requires addressing both spatial and temporal variations simultaneously. For example, it is not sufficient to analyze temporal patterns first and then spatial patterns after - a common practice in fMRI data analysis. One should not compute the correlation of a voxel with all other voxels (temporal analysis) and then use a clustering method (spatial analysis) to define a region of 'significant' activity. Instead, the data should be viewed as space-time points whose space-time trajectories must be estimated as a whole. Second, time courses are typically neither simple nor periodic; they can follow virtually any form dictated by the underlying physical processes. Finally, data are often multiparametric, with parameters influencing (i.e. coupled to) one another. For instance, in fMRI, blood flow and electrophysiology are coupled and influence each other.

The problem then becomes one of detecting the multiple modes in complex nonlinear systems. We have addressed this problem previously in our development of the EFD method, which is a probabilistic framework for estimating spatial-temporal modes of complex nonlinear systems containing multivariate interacting fields (*Frank and Galinsky, 2016a*; *Frank and Galinsky, 2016b*; *Frank et al., 2018*; *Frank et al., 2024*). These concepts are described in greater detail in Appendix 1: Entropy field decomposition. It is formally based on a field-theoretic mathematical formulation of Bayes' theorem that enables the hierarchy of multiple orders of field interactions, including coupling between fields. Its practical utility is enabled by the incorporation of the theory of *entropy spectrum pathways* (ESPs) (*Frank and Galinsky, 2014*), which uses the space-time correlations in each individual dataset to automatically select the very limited number of highly relevant field interactions. In short, it selects the configurations with maximum path entropy, summarized in the equilibrium (i.e. long time) distribution $\mu^*$. While each of these modes provides unique information on coherent spatiotemporal activity, for characterizing the total brain activity, it is often most useful and efficient to sum these modes.

A strength of the EFD method is that it uses prior information contained in individual datasets - there are no training datasets or averages across datasets - just the prior information contained within the single dataset of interest. This method has shown utility in resting-state fMRI data (*Frank and Galinsky, 2016b*) and in meteorology in the application to severe local storms, in particular tornadic supercells (*Frank et al., 2018*). The fact that this method uses prior information embedded within single datasets without the need for any 'training' is of significance to clinical studies in which important individual variations can be lost in the averaging process. It is also particularly important in the current paper where our validation necessitates comparison with single subject studies.

## Validation

Validation of any neuroimaging methods is problematic because it is not possible to directly measure brain activity at every location in the brain. Nevertheless, three methods are obvious candidates for assessment of SPECTRE's validity.

The first is comparison with fMRI, the current method of choice for whole-brain spatial localization of brain activity. However, the association of fMRI with a 'standard' for EEG is problematic because it is not measuring electrical activity, but the magnetization changes in hemoglobin as blood becomes deoxygenated during brain activity. The timescale and location of these changes can be vastly different than those produced by EEG signals. Nevertheless, its capability of spatially localizing activated brain regions merits a comparison. The most direct comparison is between fMRI and EEG data collected simultaneously, which guarantees that the brain activity measured is identical in both experiments. Such 'simultaneous fMRI/EEG' experiments are not particularly common as collecting EEG data within

an MRI scanner during imaging is notoriously difficult, and the MRI procedure significantly distorts the EEG signal. However, a recent open-source study provides such data which is sufficient for our purposes.

A more direct method for validating the ability of SPECTRE to reconstruct localized electrical activity can be constructed from intracranial EEG (iEEG) recordings collected during epilepsy studies. Such measurements consist of specially designed EEG sensors distributed linearly along a probe that is inserted deep within a brain that has been exposed by surgical removal of a portion of the skull. By selecting only these electrodes near the brain surface from the full array of electrodes, we can synthesize an artificial surface distribution of electrodes to mimic a standard noninvasive EEG experiment (albeit with a limited coverage of the brain). We have access to such data through an ongoing study which enabled this method of validation as well.

Lastly, a comparison with current 'source localization' methods would seem to be in order (*Biasiucci et al., 2019*). This comparison turns out to be the most problematic as these methods all employ a very different, and quite limited, physical model for the EEG signal and suffer from computational limitations as well. Despite attempts to make a reasonably valid comparison, it was determined that this was not possible, as described below.

## Validation with simultaneous fMRI/EEG visual task

It is notoriously difficult to get high-quality EEG data in simultaneous fMRI/EEG studies as the presence of the rapidly varying magnetic fields present in an fMRI acquisition distorts the EEG signal. However, one recent open-source simultaneous fMRI/EEG study of a well-controlled visual task (the periodic flashing checkerboard) on multiple subjects (*Telesford et al., 2023*, available from the Nathan Kline Institute) provides important data to address this question.

The fMRI procedure samples the data at relatively coarse temporal sampling and thus is most sensitive to low-frequency variations in blood oxygenation level-dependent (BOLD) activity. The most useful comparison of SPECTRE with fMRI is therefore in the lowest frequency band, $0 - 1Hz$ (for details on the fMRI acquisitions, see Visual paradigm data). The SPECTRE reconstruction in this frequency band is shown in *Figure 1* and demonstrates the ability of SPECTRE to faithfully reconstruct the spatial distribution similar to fMRI. Importantly, this comparison was performed on data from a single subject, since brain activity patterns can vary significantly between individuals, and averaging over multiple subjects obscures specific spatial variations important for validation. In the top rows of *Figure 1* is shown the fMRI EFD mode that automatically detects the activation in the primary visual cortex. In the middle row are shown the SPECTRE modes reconstructed using the 2 mm MNI anatomical atlas, chosen because it was closest in resolution ($2mm^3$) to the fMRI data ($\sim 3mm^3$). The very close correspondence between the spatial patterns is evident.

The bottom rows in *Figure 1* clearly demonstrate one of the most compelling, and perhaps surprising, aspects of SPECTRE - its ability to reconstruct activation at spatial resolution *significantly higher* than fMRI. This is a consequence of the SPECTRE reconstruction being based on the solution of the propagation of EM waves through specific tissue morphologies and bioelectric properties, provided by arbitrary resolution anatomical MRI data. The finer the resolution of the MRI scans, the more details can be available for the reconstruction. This is, of course, dependent upon the number and distribution of the EEG sensors, but certainly holds for the standard array configurations used in this paper.

Although it is an almost universally believed notion that EEG and fMRI are complementary because EEG has excellent temporal resolution but poor spatial resolution, while fMRI has poor temporal resolution but good spatial resolution, in fact, SPECTRE EEG reconstructions can achieve much higher *intrinsic* temporal *and* spatial resolution. Moreover, because there are no spatial distortions in SPECTRE, this mitigates one of the aspects of fMRI that most confounds spatial localization through signal loss and nonlinear geometric distortions. This is shown in *Figure 2*.

The slice-by-slice correlation coefficient between the activation patterns estimated by SPECTRE and fMRI is shown in *Figure 3*. Regions of very high correlation, most notable in the inferior brain regions, indicate the similarity in activation patterns detected between the two completely different neuroimaging methods (SPECTRE and fMRI). The correlations are not as strong in the superior regions of the brain, possibly due to the increased distortions in that region in this fMRI dataset. Even with perfect activation detection by both methods, the correlations would not be perfect (i.e. 1) as the two

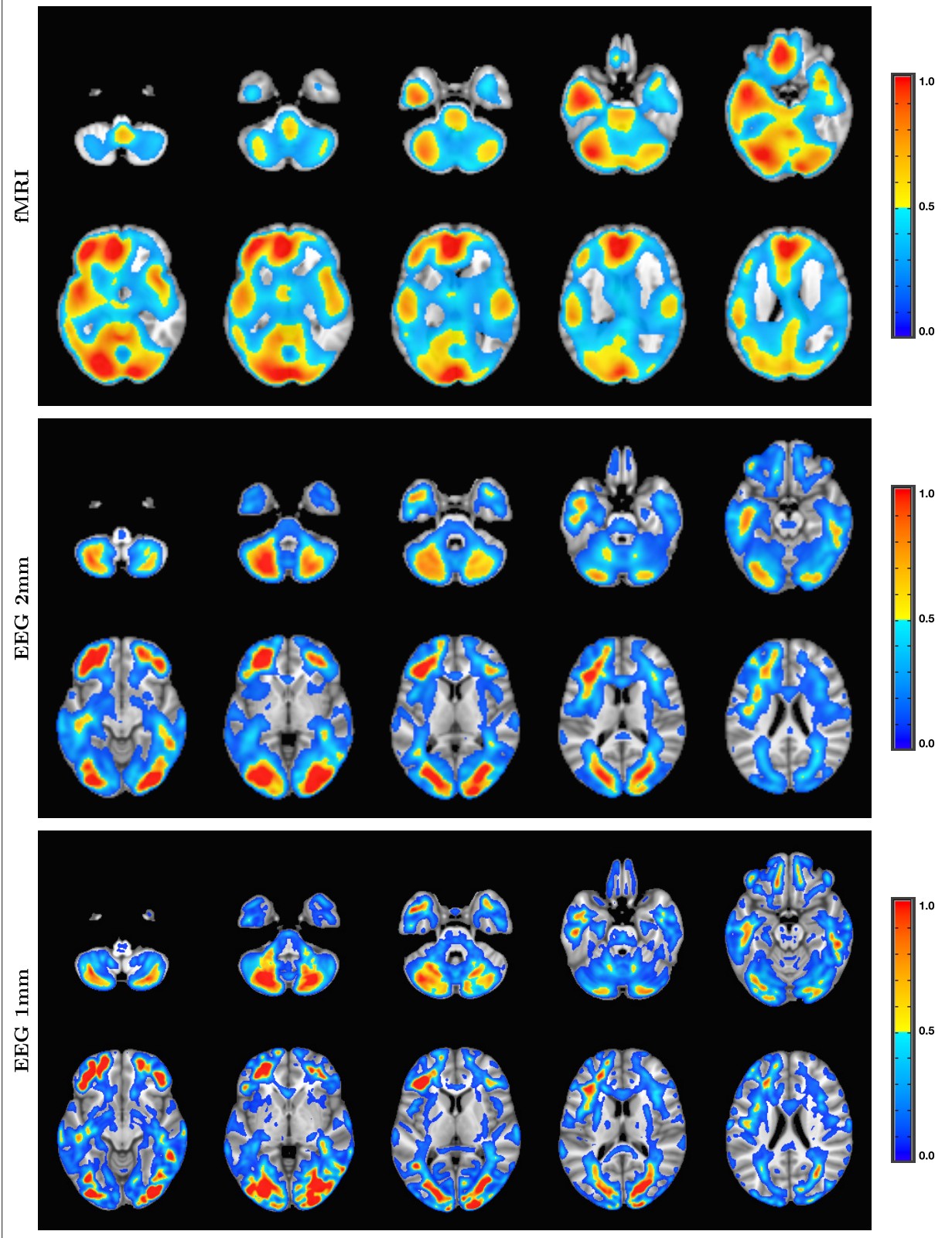

**Figure 1.** Comparison of entropy field decomposition (EFD) reconstructed functional MRI (fMRI) activity (top) with SPatially resolved EEG Constrained with Tissue properties by Regularized Entropy (SPECTRE) electroencephalography (EEG) reconstruction in the frequency band 0–1 Hz at both 2 mm (middle) and 1 mm (bottom) spatial resolution (axial view) from a single representative subject from an open-source study with simultaneous fMRI and EEG (***Telesford et al., 2023***). In both cases, the weighted sum of the power over all modes is shown. The task was a simple 8 Hz flashing checkerboard

*Figure 1 continued on next page*

*Figure 1 continued*

with 4 on/off cycles. The nonlinear registration of the fMRI to the anatomical template in the fMRI data (top) is imperfect because of significant field-induced nonlinear geometric distortions in the fMRI data. The colors are the weighted sum over all estimated amplitudes of the activation modes. Intensities are scaled between 0 and 1, and threshold at 0.6.

methods are measuring different physical processes. However, the smooth variations are indicative of nonrandom correlations between two vastly different imaging modalities.

It should be noted that the 'simple' periodic flickering checkerboard stimulus not only activates the primary visual cortex but activates other visual and supplementary fields as well, as is evident from the activity patterns in *Figure 1*. A simple stimulus does not imply a simple activation pattern. This notion was a primary motivation for our development of the EFD method for fMRI (*Frank and Galinsky, 2016b*). The activation mode reconstructions for both the fMRI and SPECTRE data are based on the EFD, which detects complex nonlinear interacting spatial-temporal modes of activity (*Frank and Galinsky, 2016b*). Thus, although the task is a 'simple' visual stimulation, our analysis is not expected to simply detect activity in only the visual cortex, as would be produced by a more standard

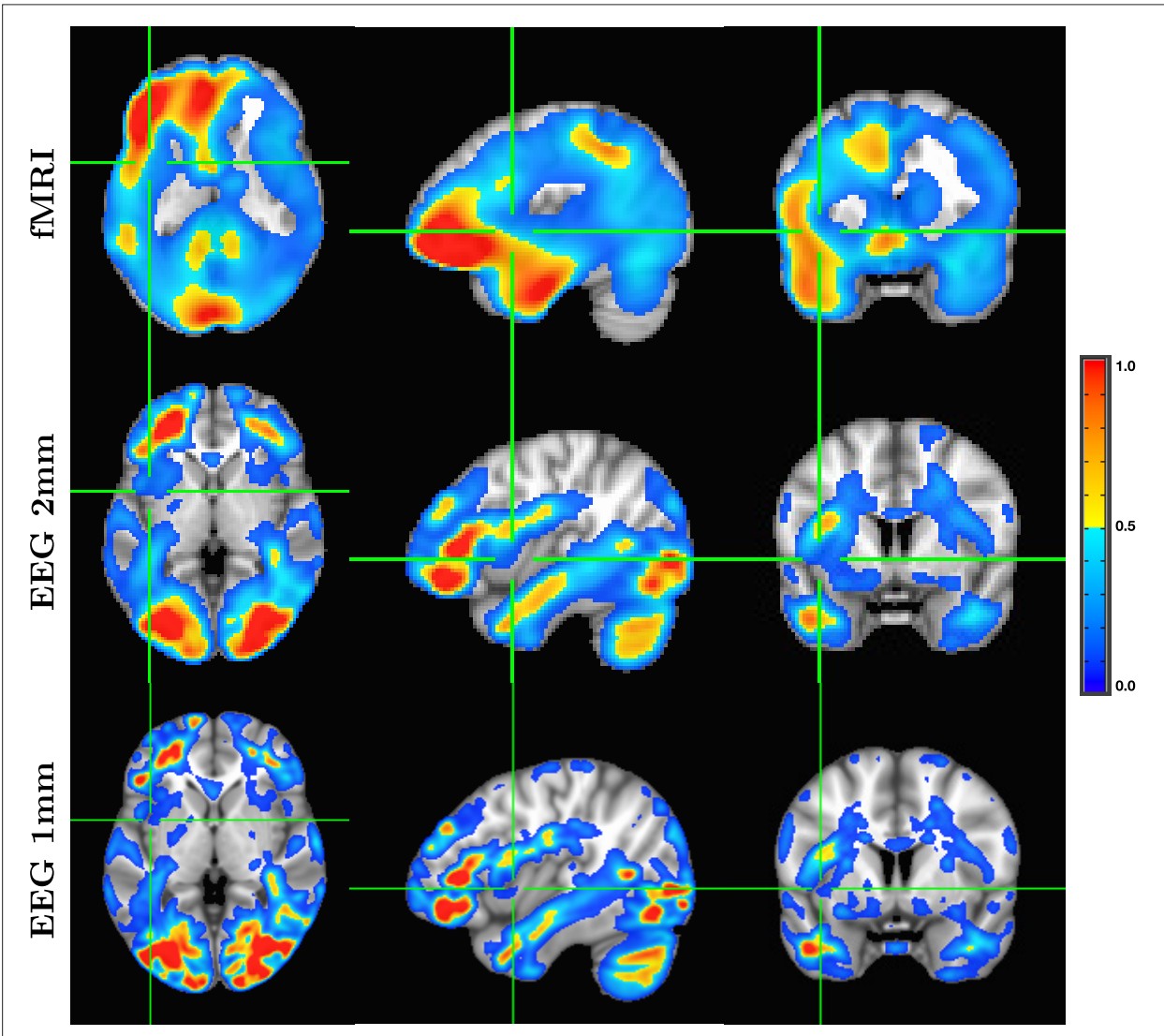

**Figure 2.** A detailed visualization of three orthogonal views of data in *Figure 1* demonstrating the fine spatial resolution produced by SPatially resolved EEG Constrained with Tissue properties by Regularized Entropy (SPECTRE), and the ability to reconstruct activations in regions prone to severe distortions in functional MRI (fMRI), such as the frontal lobes and cerebellum. The colors are the weighted sum over all estimated amplitudes of the activation modes. Intensities are scaled between 0 and 1, and threshold at 0.6.

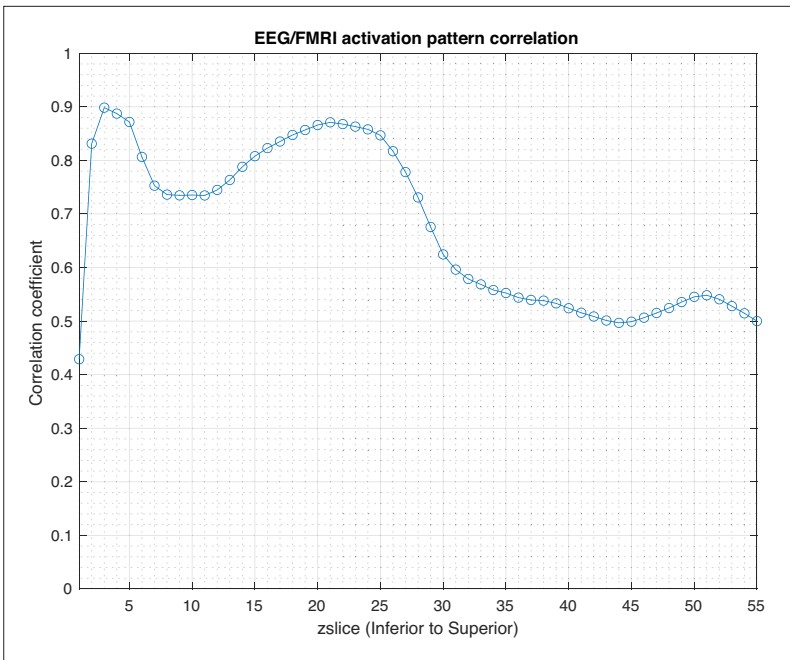

**Figure 3.** Correlation coefficient in each axial slice (from inferior to superior) between the activation patterns estimated by SPatially resolved EEG Constrained with Tissue properties by Regularized Entropy (SPECTRE) electroencephalography (EEG) and the functional MRI (fMRI) for the data in *Figure 1*. Regions of high correlation indicate the similarity in activation patterns detected between the two completely different neuroimaging methods (SPECTRE and fMRI). Reduction of the correlations in the superior regions of the brain, possibly due to the increased distortions in that region in this fMRI dataset.

regression approach (*Telesford et al., 2023*), but in a more complex set of brain networks. Indeed, multiple EFD modes are produced, though we have only shown the one incorporating the primary visual cortex. As we have argued previously (*Frank and Galinsky, 2016b*), EFD analysis is more sensitive than simple regression techniques to the complex brain activation patterns predicted by neuroscience, and less sensitive to erroneous identification of noise or nonindependent modes than the independent component analysis (ICA) (*Frank and Galinsky, 2016b*). Indeed, one of our observations from both the fMRI and EEG data used in this study (*Telesford et al., 2023*) is the appearance of PFC activations associated with visual stimulation, which has been suggestive of conscious visual perception (*Libedinsky and Livingstone, 2011*; *Paneri and Gregoriou, 2017*). Addressing this question is beyond the scope of the current paper.

## Validation with simultaneous fMRI/EEG attention paradigm

Simultaneous EEG/fMRI were collected from subjects within a standard clinical 3T MRI scanner (see Attention paradigm data for details). The stimuli and paradigm are described in detail in *Grinband et al., 2017*. Briefly, bimodal stimuli consisting of short ($\sim 1s$) streams of simple tones (600 and 1000 Hz) alternating at 10 Hz were delivered concurrently with phase-reversing (6 Hz) checkerboard patterns presented at fixation. Participants were instructed to selectively attend to either the visual or auditory aspect of the bimodal stimulus and respond when the stream of stimuli in the attended modality ends.

SPECTRE processing was performed in the alpha band. The appearance of visual stimuli elicited a reduction of ongoing alpha (7–14 Hz) activity ('event-related desynchronization' [ERD]) over occipital cortex, believed to occur when cortical regions are brought 'online' for information processing (*Klimesch, 2012*). As in previous studies, e.g., *Foxe and Snyder, 2011*, attended visual stimuli elicited increased (more negative) amplitude of the alpha ERD compared to unattended stimuli (*Figure 4A and B*). In contrast, unattended, compared to attended, visual stimuli elicited a greater reduction in ongoing spectral activity within the 5–15 Hz frequency range over bilateral middle frontal cortex (*Figure 4C and D*). We estimated the neural sources of these attention-related modulations of

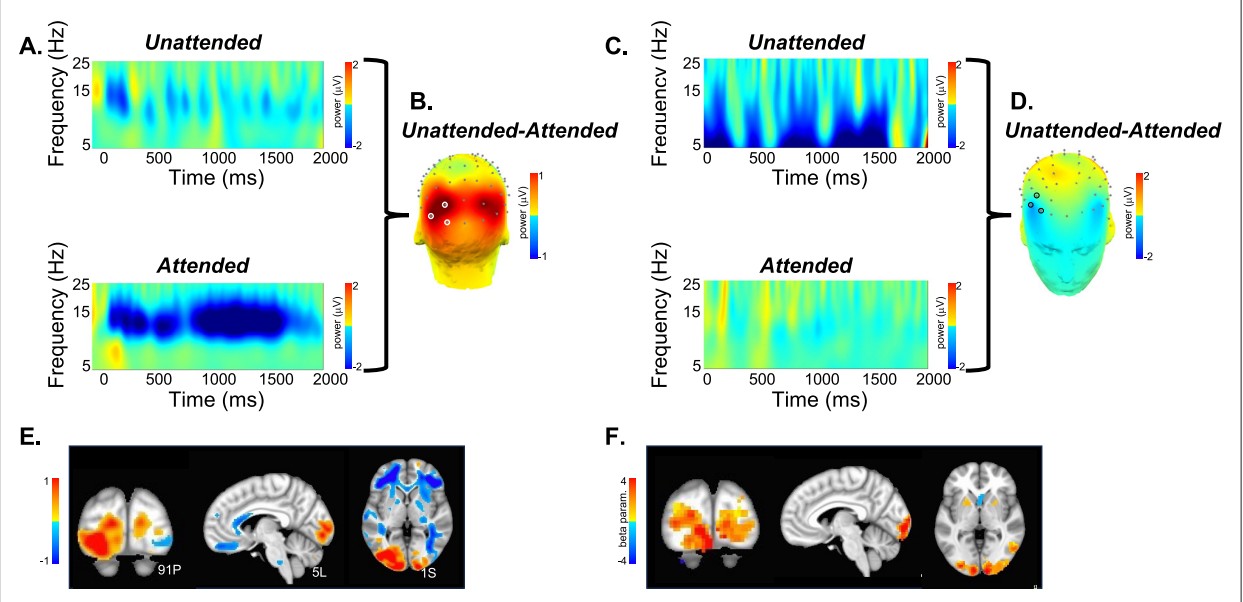

**Figure 4.** Validation of SPECTRE against standard EEG spatial and frequency mapping and simultaneously acquired fMRI. (**A**) Baseline-corrected electroencephalography (EEG) activity from a single subject elicited by unattended (top) and attended (bottom) visual stimuli averaged across the cluster of three occipital electrode sites (PO7, PO3, O1) denoted in **B** by white circles. Over the broad alpha frequency band (7–16 Hz), there was a reduction in total power (from the pre- to poststimulus latency interval) which was greater for attended, compared to unattended, visual stimuli. (B) Scalp topography of the mean difference in oscillatory (8–12 Hz) activity for unattended minus attended visual stimuli across the 0–2000 ms latency interval. As expected, attention modulated (reduced) the power of these oscillations over the visual cortex. (**C**) As in A for three frontal electrode sites (F6, F8, AF6) denoted in D by black circles. In contrast to visual cortex, in bilateral frontal regions, unattended visual stimuli elicited a greater reduction of oscillatory activity between 5 and 10 Hz (theta-alpha frequency). (**D**) Frontal view of the unattended minus attended difference topography between 0 and 2000 ms in the 8–12 Hz frequency band. (**E**) SPatially resolved EEG Constrained with Tissue properties by Regularized Entropy (SPECTRE) Power estimates derived from mean (baseline-corrected) oscillatory power between 0 and 2000 ms and across 8–12 Hz for the same subject shown in panels **A–D**, superimposed on the MRI Montreal Neurological Institute (MNI) template brain. Hot colors (yellow to red) indicate greater attention-related modulation (reduction) of activity, and the inverse for warm colors (light to dark blue). (**F**) Blood oxygenation level-dependent (BOLD) signal (beta parameter estimate) contrasting activation to visual stimuli when attended vs activation to the same stimulus when unattended. Attention-related enhancement of the BOLD signal in visual cortex mirrors the reduction in alpha power obtained in the same subject using EEG.

oscillatory activity across the 8–12 Hz frequency band, which encompassed both the occipital and frontal activities (*Figure 4E*). Their anatomical localization was remarkably consistent across several individuals (*Figure 5*).

A direct comparison of the activation maps derived from both fMRI and EEG using SPECTRE for a single study within two subjects (i.e. without any average over studies or subjects) is shown in

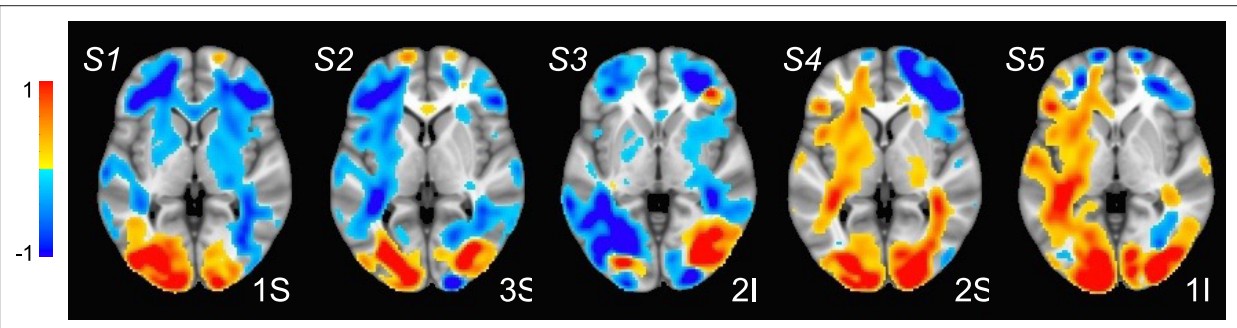

**Figure 5.** Estimated localization of neural activity for 8–12 Hz oscillatory activity (unattended minus attended; 0–2000 ms) for five participants (S1-S5). Colors are as in *Figure 4E*. A prominent bilateral occipital source associated with increased attentional modulation is observable in all participants. A bilateral source localized in the middle frontal cortex and indicating less modulation is also consistently observed across participants. Note that these are difference maps from the weighted sum over all estimated amplitudes of the activation modes, so that the intensities are scaled between –1 and 1, and threshold at absolute value 0.6.

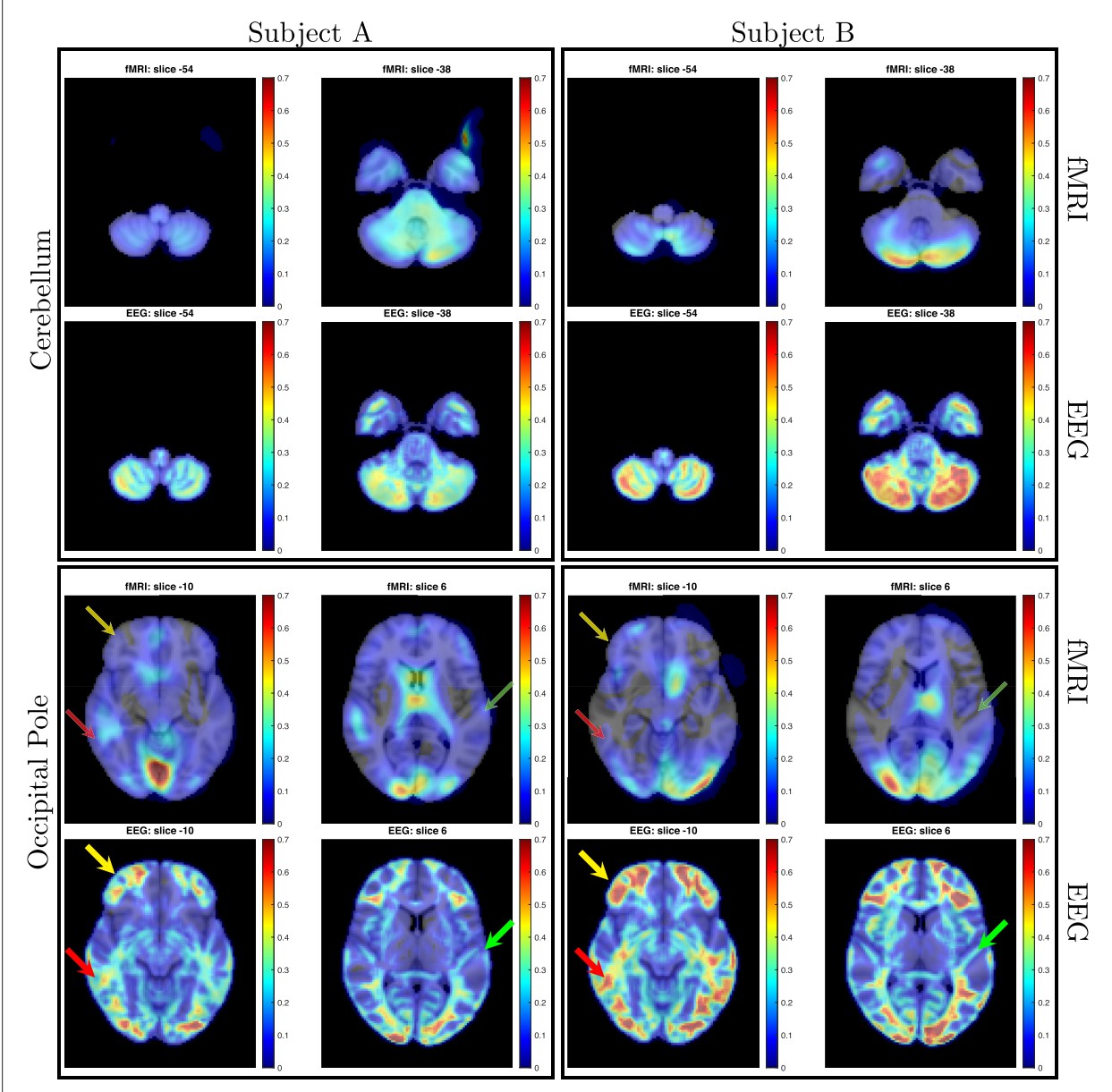

**Figure 6.** Direct comparison of activation maps from two participants (Subject A, left; Subject B, right) in the bimodal (auditory + visual) stimulation paradigm described for *Figures 3 and 4*. In each subject, two brain regions - the cerebellum and the occipital pole (top and bottom rows, respectively) - were delineated based on the MRI Montreal Neurological Institute (MNI) atlas, and entropy field decomposition (EFD) activation maps were correlated across these entire regions. Correlation coefficients were as follows: for Subject A, cerebellum=0.74, occipital pole=0.70; for Subject B, cerebellum=0.70, occipital pole=0.84. Correlations were computed only for regions exhibiting activation levels above 0.1. In contrast to functional MRI (fMRI), the SPatially resolved EEG Constrained with Tissue properties by Regularized Entropy (SPECTRE) technique identified robust activations in bilateral middle and inferior frontal cortex (indicated by yellow arrows) and middle temporal cortex (red arrows). It also discerned activations along the superior temporal cortex, including areas encompassing the primary auditory cortex (green arrows).

*Figure 6*. The comparison is made by choosing specific regions of interest defined in the MNI atlas (occipital cortex and cerebellum) and correlating the activation maps derived from EFD for fMRI and SPECTRE from EEG. Comparison of the similarity of activated regions in individual subjects is generally a nontrivial problem. This is particularly true in the current case where the spatial distortions in fMRI (and lack of them in SPECTRE) make measures such as mean-squared error difficult to interpret. Therefore, the computation of the correlation coefficient over a predefined atlas ROI is a reasonable conservative measure of statistical significance.

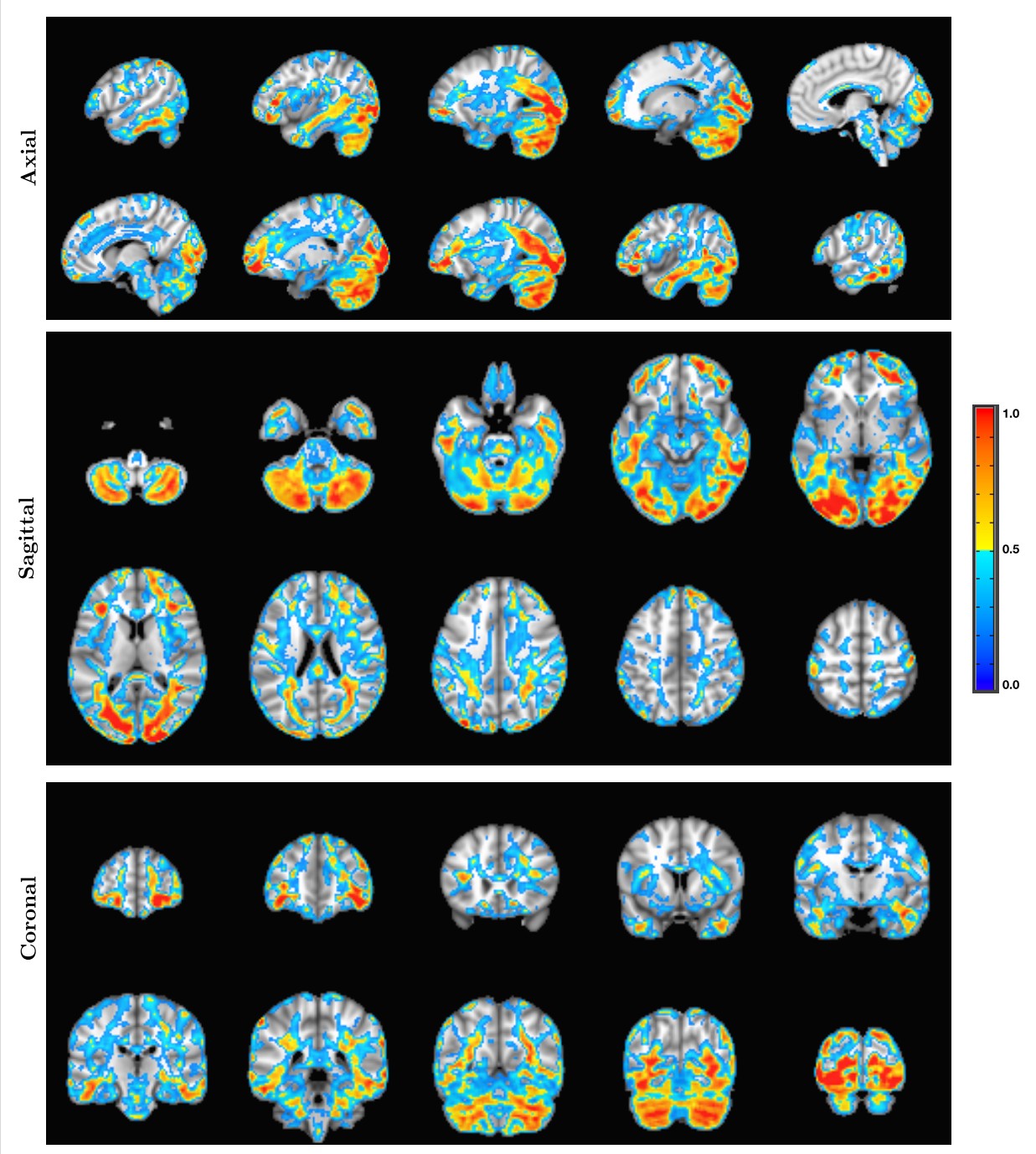

**Figure 7.** Orthogonal slices from whole-brain electric field activation maps from a 2 mm SPatially resolved EEG Constrained with Tissue properties by Regularized Entropy (SPECTRE) reconstruction of electroencephalography (EEG) data from a single subject in the attention study. The colors are the weighted sum over all estimated amplitudes of the activation modes.

As an example of the type of whole-brain electric field activation maps that are possible with SPECTRE is shown in the montage of orthogonal slices from a 2 mm reconstruction, *Figure 7*, from one of the subjects of this same attention study.

## Statistical significance of simultaneous EEG/fMRI results

Direct comparison of activation maps from two participants in the bimodal (auditory + visual) stimulation paradigm described for *Figures 3 and 4* is shown in *Figure 6*. In each subject, two brain regions

- the cerebellum and the occipital pole (top and bottom rows, respectively) - were delineated based on the MNI atlas, and EFD activation maps were correlated across these entire regions. Correlation coefficients were as follows: for Subject A, cerebellum = 0.74, occipital pole = 0.70; for Subject B, cerebellum = 0.70, occipital pole = 0.84. Correlations were computed only for regions exhibiting activation levels above 0.1. In contrast to fMRI, the SPECTRE technique identified robust activations in bilateral middle and inferior frontal cortex (indicated by yellow arrows) and middle temporal cortex (red arrows). It also discerned activations along the superior temporal cortex, including areas encompassing the primary auditory cortex (green arrows). We emphasize that the difference in activation maps provided by SPECTRE and fMRI is not expected to be identical, as EEG and fMRI are not measuring the same physical quantities. Indeed, fMRI is measuring rather poor proxies of the brain electrical fields. Therefore, it would be remarkable if these two methods did not have significant differences. *Figure 4* provides the best example, where our EEG method showing deactivation (blue in E) is consistent with what can be considered a gold standard for EEG - the direct surface recordings near the scalp where the deactivation in the frontal lobes (C, D in blue) corresponds. It should also be noted that this is not simply a visual task, but an attention task, for which these activation patterns are well known, and thus provides yet another form of validation.

Therefore, the high correlation coefficients between the maps, *Figure 6*, are therefore indicative of the consistency between the fMRI and SPECTRE results in the ROI. Note that this does *not* imply similarity over the entire region shown. Indeed, the SPECTRE results show enhanced sensitivity to activation in regions not seen in the fMRI.

## Validation through iEEG recordings: surface electrode vs complete electrode array

While comparison with fMRI can validate the correct detection of activated brain regions and networks, as shown in the previous section, it cannot inform the question of correct detection of electrical signals, since fMRI is based on a completely different contrast mechanism related to blood oxygenation. A direct validation of SPECTRE's ability to faithfully reconstruct deep EM activity is, to our knowledge, only achievable with one type of data: iEEG recordings such as those used in medically refractory epilepsy patients for seizure onset localization where the electrodes are known to be adjacent to the site of electrical activity (*Ramantani et al., 2016*; *van Mierlo et al., 2020*). We analyzed an iEEG recording of a seizure localized in the left medial temporal region acquired at Northwell Health, NY. All implanted electrodes are shown in *Figure 8* (top row) with each yellow dot depicting one recording contact. Comparing the SPECTRE reconstruction using all of the sensor data with one using only a subset of the data comprised of only the sensors on the surface of the brain (red dots in *Figure 8*, top row) allows the quantitative assessment of how closely the results from a set of surface electrodes correspond to those produced by intracranial measurements recording signal very close to the sources. The results are shown for the alpha frequency band in *Figure 8* and reveal a very close correspondence between the SPECTRE mode reconstruction. Results for Subjects 2–4 are shown in *Appendix 2—figures 1–3* and show similar agreement.

## Validation through iEEG recordings: comparison with simulation

A traditional approach for validating estimation methods is to compare results against a 'ground truth' derived from numerically simulated signals. This is a standard procedure in source reconstruction methods, where simulated point dipole sources are embedded within a brain model (e.g. a high-resolution MRI scan). The forward problem is then solved to generate the dipole fields at the brain surface, and the resulting simulated signals are used to estimate the original source locations. While this approach works well for dipole-based models of brain activity, it is not directly applicable to the more realistic WETCOW model of brain waves.

However, an alternative validation strategy is possible by comparing SPECTRE data to the most reliable ground-truth data available: iEEG. One of the most striking predictions of the WETCOW theory is the presence of coherent, sustained cortical wave loops - a phenomenon demonstrated through numerical simulations in a realistic brain model derived from HRA data (Figure 10, top). This prediction provides a natural benchmark for validating the SPECTRE method.

An application of SPECTRE to a WETCOW analysis of iEEG recording of several epileptic seizure onsets in insular posterior opercular and in hippocampus areas that provides our first experimental

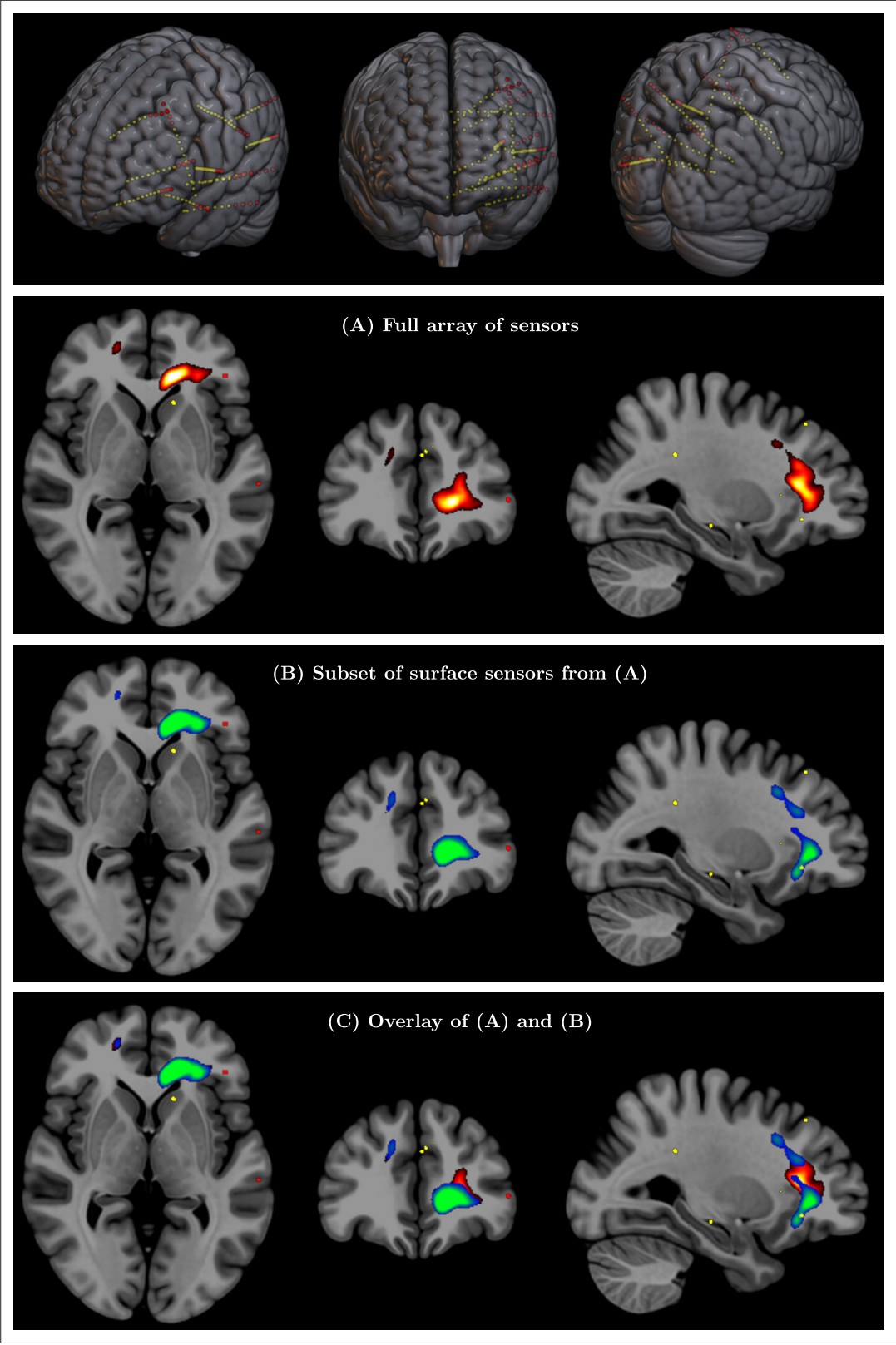

**Figure 8.** (Top row) Full array of intracranial electroencephalography (EEG) contacts from a recording in a medically refractory epilepsy patient (yellow dots). Red dots indicate a subset of surface-only electrodes to mimic a standard noninvasive (i.e. extracranial) EEG study. SPatially resolved EEG Constrained with Tissue properties by Regularized Entropy (SPECTRE) alpha band reconstruction from (**A**) full array of intracranial EEG sensors from an

*Figure 8 continued on next page*

*Figure 8 continued*

epilepsy study (yellow dots) in top row and (**B**) from subset of surface electrodes (red dots) in top figure. (**C**) Overlay of (**A**) and (**B**) validating that the surface-based is correctly reconstructing the local electric field potential detected by the intracranial electrodes.

evidence of the existence of our hypothesized cortical loops is shown in Figure 10 (bottom right). SPECTRE reconstruction confirms the existence of cortical wave loops in Figure 10 (bottom right), consistent with the numerical simulations shown in Figure 10 (top). The intracranial leads are shown in Figure 10 (bottom left).

## Statistical significance of iEEG results

A deep-surface-full comparison of modes for seizures datasets was run for all 5 iEEG subjects with 44 events total. The correlation plot shown in *Figure 9* demonstrates that the correlations are very high. We ran three t-tests on Fisher's Z-transformed correlation values (full-deep/full-surface, full-deep/deep-surface, full-surface/deep-surface), and the t-tests show that full-deep/full-surface correlations are very similar (null hypothesis is not rejected, $p = 0.2368$, $t(43) = 1.19$, $p > 0.001$, 95% CI [–0.0370, 0.1457], SD = 0.3004), but the full-deep/deep-surface and full-surface/deep-surface t-tests show statistically significant differences ($p = 1e-8$, $t(43) = 7.05$, $p > 0.001$, 95% CI [0.1827, 0.3291], SD = 0.2408, and $p = 8e-6$, $t(43) = 5.06$, $p > 0.001$, 95% CI [0.1213, 0.2818], SD = 0.2640). These results support the claim that the SPECTRE reconstruction of the spatial distribution of deep electrical activity from the surface measurements accurately reflects the true spatial localization of the deep electric fields.

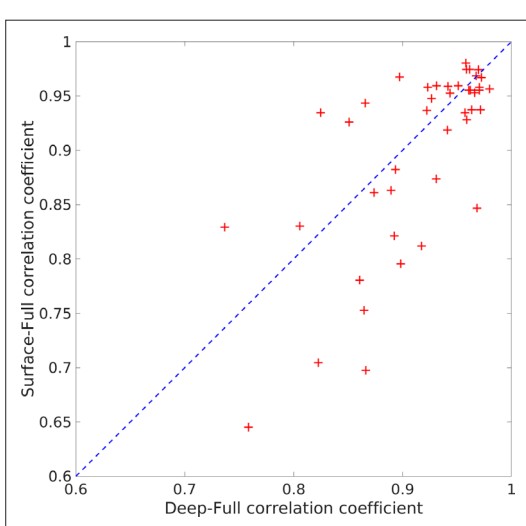

**Figure 9.** Statistical comparison of full vs surface intracranial electroencephalography (EEG) estimates. The horizontal axis represents the correlation coefficients between the estimates obtained from the full set of electrodes and the deep electrodes (adjacent to the source). The vertical axis represents the correlation coefficients between the full set of electrodes and just the surface electrodes, as would be collected in a standard (extracranial) EEG experiment. The results are highly correlated and thus support the claim that the SPatially resolved EEG Constrained with Tissue properties by Regularized Entropy (SPECTRE) reconstruction of the spatial distribution of deep electrical activity from the surface measurements accurately reflects the true spatial localization of the deep electric fields.

## Investigation of the 'reward circuit'

Having validated the SPECTRE method directly with simultaneous fMRI/EEG, iEEG, and an attention paradigm, we investigated the ability of SPECTRE to faithfully reconstruct the well-known neural 'reward circuit' that is one of the most important in understanding human cognition, emotion, and behavior (*Schultz, 2015*; *Haber, 2017*; *Banich and Floresco, 2019*) and is of great clinical significance in the understanding of addiction (*Koob and Roberts, 1999*; *Gardner, 2011*), mood disorders (*Naranjo et al., 2001*; *Russo and Nestler, 2013*), and a variety of other conditions (*Lewis et al., 2021*; *Figure 10*).

We demonstrate that SPECTRE using standard EEG data can accurately map human reward pathways akin to results previously only seen via fMRI. Indeed, fMRI results have highlighted a reward system within the brain that includes midbrain dopamine-producing regions (the substantia nigra pars compacta, the ventral tegmental area), the ventral striatum, and multiple regions within the human PFC (*McClure et al., 2004*). Other research using fMRI and source localization of EEG data suggests that the anterior cingulate cortex also plays a key role in reward processing (*Holroyd and Coles, 2008*). In a unifying theory, it has been proposed that all the aforementioned regions work together as a neural system for the

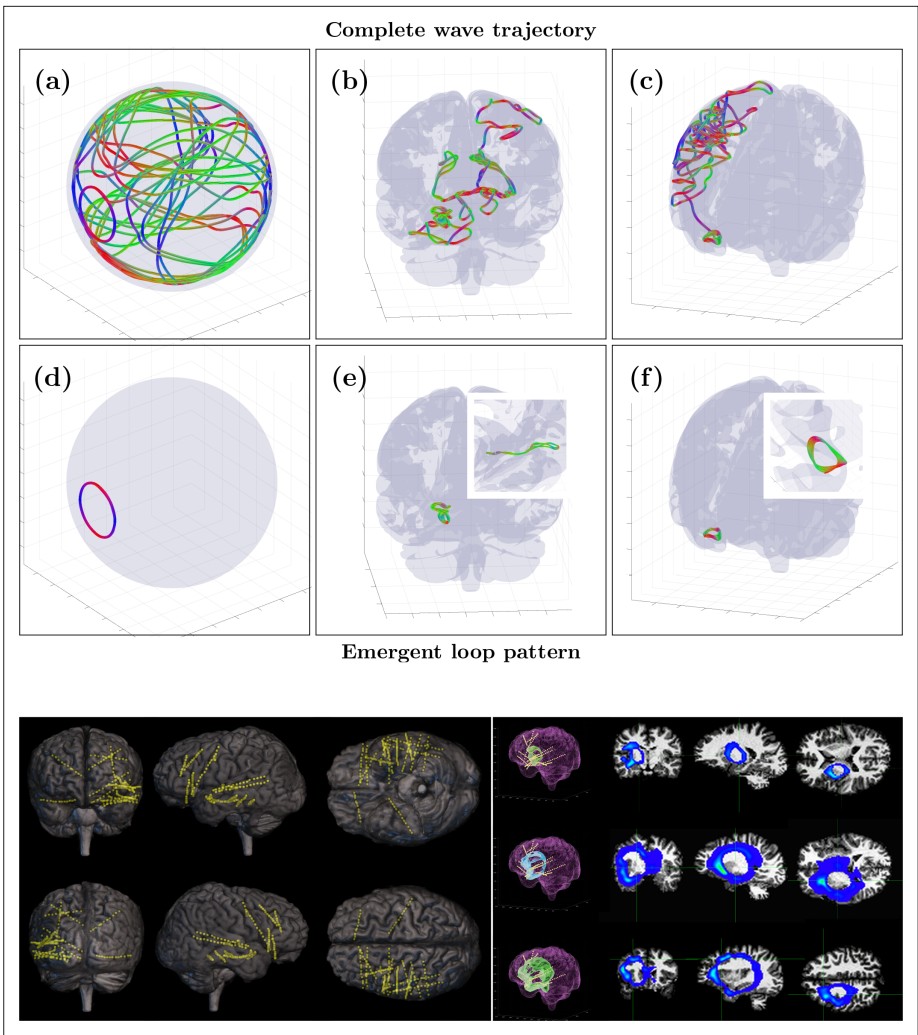

**Figure 10.** Validation of weakly evanescent transverse cortical waves (WETCOW) model with intracranial measurements and SPatially resolved EEG Constrained with Tissue properties by Regularized Entropy (SPECTRE) reconstruction. (Top) Examples of wave trajectories obtained in simulation of wave propagation in real data cortical fold tissue model. Panels (**a, c**) show the complete trajectories, and panels (**d, e**) show the emergent stable wave loops. The spherical cortex shell model is used for panels (a) and (d), and the cortical fold model is used for panels (**b**), (**c**), (**e**), and (**f**) (reprinted from *Galinsky and Frank, 2020a*, 2020 Massachusetts Institute of Technology. All rights reserved). The colors encode wave propagation: red - left/right, green - anterior/posterior and blue - dorsal/ventral. (Bottom) (left) EEG contacts and (right) detected WETCOW cortical loops from intracranial electroencephalography (iEEG) recordings of epileptic seizure onset in insular posterior opercular area.

optimization of reward-driven behavioral change (i.e. reinforcement learning; *Holroyd and Coles, 2002*).

This is of particular clinical significance because addictive behaviors have long been known to be subserved by specific brain regions operating in concert as the reward circuit (*Koob and Volkow, 2010*; *Leshner, 1997*; *Tapert et al., 2003*; *Tyree and de Lecea, 2017*; *Kallen et al., 2023*). The reward circuit is involved in processing rewarding stimuli of any sort, and, in drug addiction, substances of abuse (e.g. amphetamine) increase dopamine release in a protracted and less regulated manner as compared to typical stimuli, resulting in synaptic plasticity and altered functioning of this circuit over time.

For our analysis, we used a large gambling task dataset that includes 500 participants available for download from https://osf.io/65x4v/. The details of the dataset and an extensive analysis using standard EEG analysis methodologies are presented in *Williams et al., 2021*. The relevant information from this study is presented in Appendix 2: Reward Circuit Data.

For each subject trial $n = 10$ power modes were calculated and summed to form the single space-time SPECTRE mode (see Mode reconstruction). *Figure 11* shows three orthogonal slices of the difference in EFD power summed over all modes between conditions, averaged over all subjects. Activation in key regions of the reward circuit, including the frontal lobes, anterior cingulate gyrus, accumbens, and amygdala, is clearly evident. Strong negative activation (i.e. deactivation) is evident in several structures, including the supplementary motor cortex and the parietal operculum cortex. Activation is also apparent in the lingual gyrus and around the calcarine fissure and, as expected, in bilateral subcortical structures.

In *Figure 12* is shown the power per brain regions as defined by the Harvard-Oxford 2 mm cortical (top) and subcortical (bottom) atlases. In the cortical regions (top), strong activation is apparent in the frontal cortex (medial, orbital, operculum), cingulate gyrus, paracingulate gyrus, and insular cortex. Activation in the accumbens is apparent from the data in the subcortical atlas *Figure 12* (bottom). These activated regions are consistent with the known elements of the human brain reward circuitry.

Images of statistical significance ($p < 0.0001$) are shown in *Figure 13*. It should be noted that the determination of statistical significance with SPECTRE by 'traditional' methods is potentially misleading as they will tend to *underestimate* activation significance. The estimation of the modes in SPECTRE employs EFD (*Frank and Galinsky, 2016a*; *Frank and Galinsky, 2016b*), which is a probabilistic formulation that *by construction* incorporates space-time neighborhood connectivity so that spatially and temporally coherent patterns ('clusters') are more probable. Traditional methods have the option for 'clustering' regions of activation post hoc into their general class of techniques called 'bootstrapping' or 'permutation inference'. Cluster post-detection of an activation is incommensurate with our view of the estimation process, wherein the clustering in space-time is a key component indicator of high-probability regions of space-time. Spatially and temporally coherent patterns may be of low amplitude with apparent low significance by traditional means, but those intensities are within a mode that contains very high significance in cortical regions (e.g. *Figure 13*), which is predicted by the WETCOW model.

## Statistical significance of reward circuit results

Mass univariate voxel-wise statistical analysis across the whole brain was performed using AFNI 3dttest++. The first level fixed effects were analyzed to produce contrast estimates computing the mean activation for each condition (the SPECTRE power modes obtained for pre- and poststimulus reward experiment). Statistical significance (t-statistic) between the SPECTRE power modes pre- and poststimulus reward experiment. Calculations of statistical significance (two-sample t-statistic) between the SPECTRE power modes pre- and poststimulus reward experiment were performed using the standard AFNI 3dttest++ algorithm. Significance threshold was $p = 1e-8$, indicating strong statistical significance. The permutation/randomization multiple comparisons correction method was used to control the family-wise error rate and false discovery rate with AFNI's 3dttest++ cluster-level thresholding through the -ClustSim option of the AFNI 3dttest++ algorithm. Images of statistical significance ($p < 0.0001$) are shown in *Figure 13*. These results support the claim that SPECTRE can reliably reconstruct whole-brain electric field activity.

## Comparison with state-of-the-art source localization methods

There is a long history of attempts to spatially localize EEG activity, and these are generally called 'source localization' or 'source reconstruction' methods (*Pascual-Marqui et al., 1994*; *Hallez et al., 2005*; *Hallez et al., 2007*; *Dattola et al., 2020*). These methods are fundamentally different from the SPECTRE approach as they are solving a different problem than SPECTRE (as described in A new physical theory of brain waves) that involves numerous stringent assumptions about brain electrical activity such as a fixed set of static dipole sources, an idealized geometric model of the head reduced to a few (typically three) shells, that spatially close points are more likely synchronized and the smoothness of the solution (see *Taulu and Larson, 2021*, and references therein).

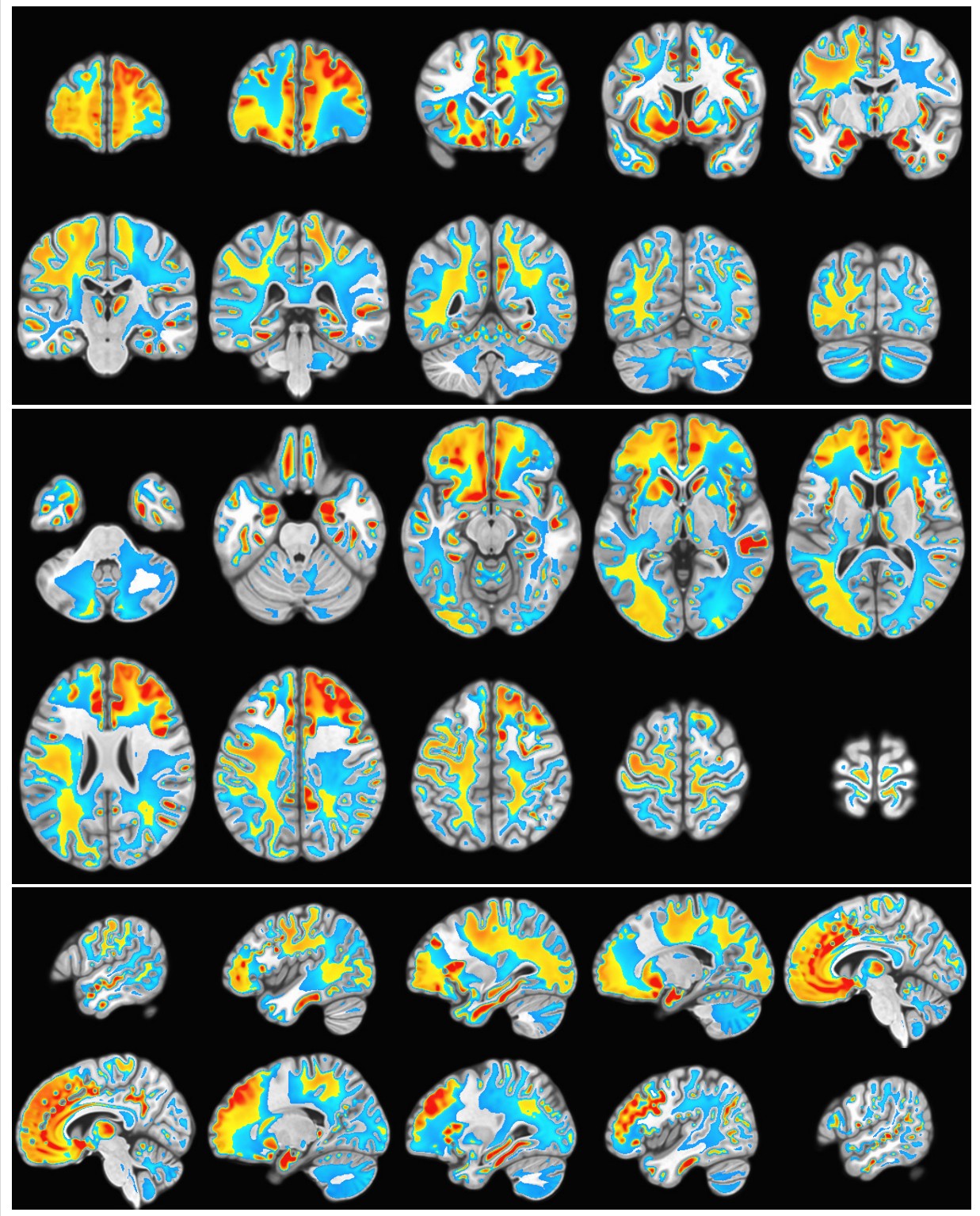

**Figure 11.** Gambling task electroencephalography (EEG) from 500 subject cohort. Alpha power of the weighted sum over the first *n* = 10 SPatially resolved EEG Constrained with Tissue properties by Regularized Entropy (SPECTRE) modes. Activation in key regions of the reward circuit, including the frontal lobes, paracingulate gyrus, accumbens, and amygdala, is clearly evident. Negative activation (i.e. deactivation) is evident in the supplementary motor cortex and the left temporal-parietal regions.

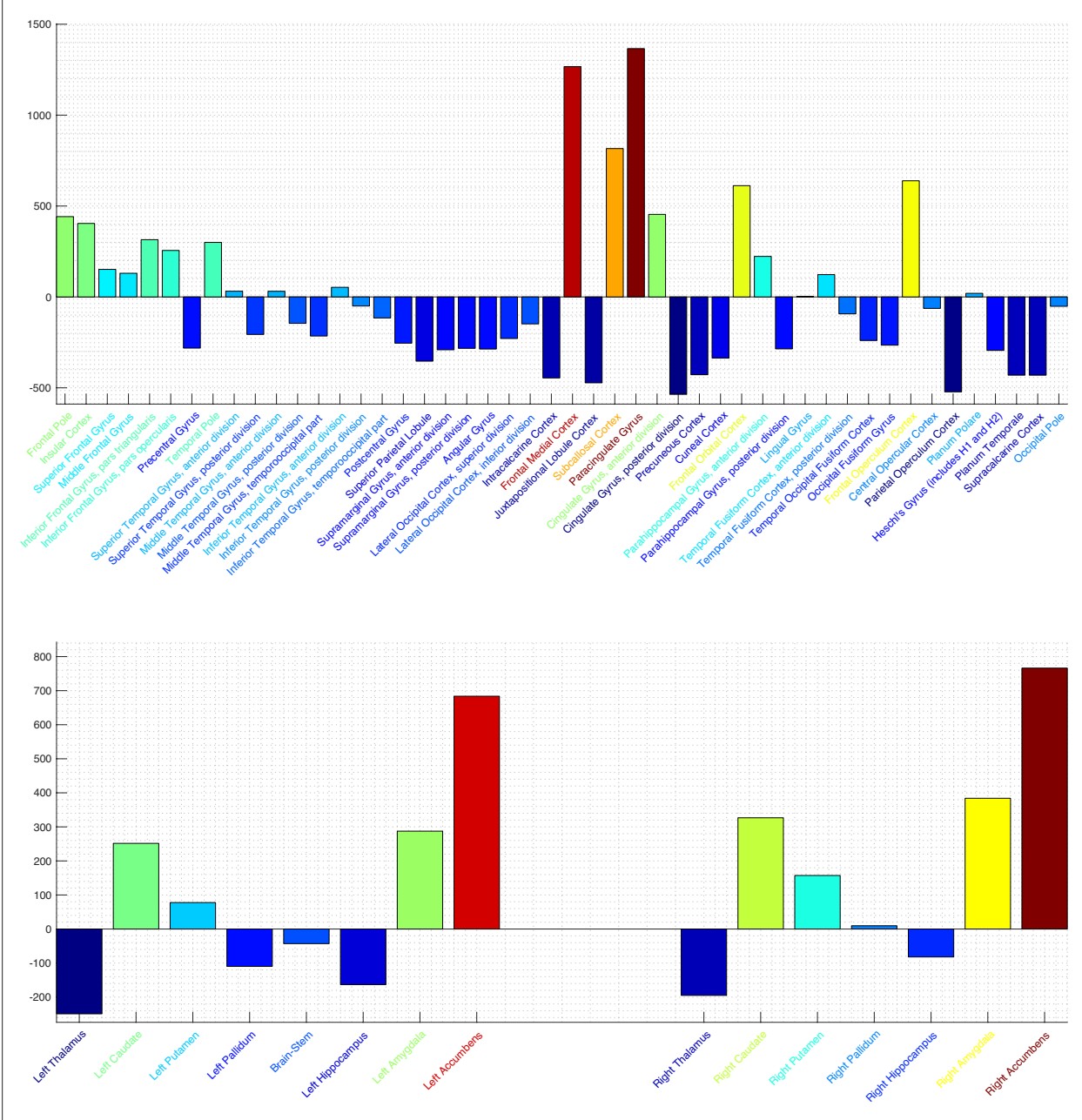

**Figure 12.** SPatially resolved EEG Constrained with Tissue properties by Regularized Entropy (SPECTRE) power per brain region in the Harvard-Oxford 2 mm cortical (top) and subcortical (bottom) atlases. Colormap is from hot/yellow (activated) to blue (deactivated). Activation in key regions of the reward circuit, including the frontal lobes, paracingulate gyrus, subcallosal cortex/nucleus accumbens, and amygdala, is clearly evident. Negative activation (i.e. deactivation) is evident in the supplementary motor area, posterior cingulate, and thalamus. Activation of the important reward element accumbens is evident in the bottom plot. Also of note is the relatively similar activation in the bilateral subcortical elements.

These methods all implicitly assume the 'quasi-static' approximation to the EM field equation, which entails ignoring the time-dependent terms in Maxwell's equations, which are dependent on tissue conductivity properties which are themselves frequency dependent. The resulting solutions are therefore static, have no frequency dependence, and are insensitive to the detailed spatially variable electrical properties of the tissues. However, as discussed in detail in the development of the WETCOW model (*Galinsky and Frank, 2020b*; *Galinsky and Frank, 2021*), these assumptions are incompatible with the basic physics of brain electrical activity. The SPECTRE approach is to employ the WETCOW model and solve the actual physical problem of the complete Maxwell's equations in

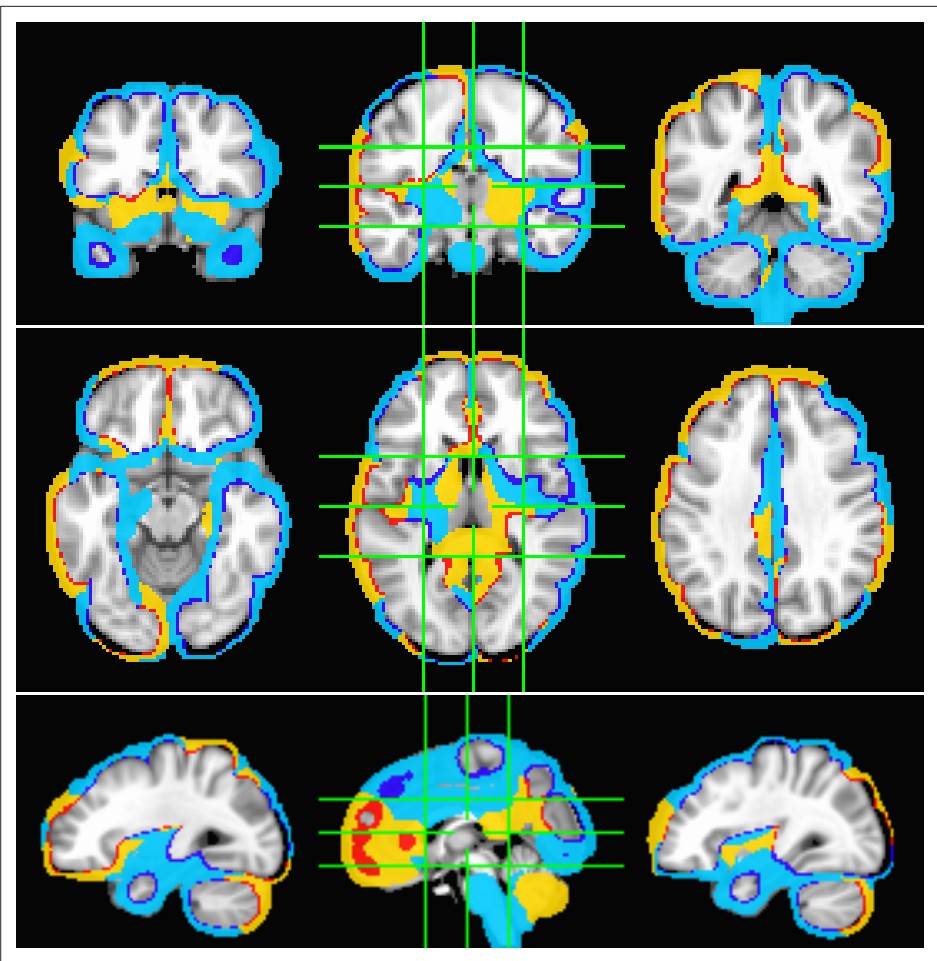

**Figure 13.** Statistical significance. t-Statistic between the SPatially resolved EEG Constrained with Tissue properties by Regularized Entropy (SPECTRE) power modes pre- and poststimulus reward experiment. Calculations were performed using the standard AFNI 3dttest++ algorithm. Yellow/red color reflects positive changes, blue color reflects negative changes. Significance threshold was $p=10^{-8}$, indicating strong statistical significance.

an inhomogeneous and anisotropic medium. It is specifically these dependencies that give rise to the previously undiscovered WETCOW waves that propagate preferentially along the gradients of local tissue inhomogeneity and anisotropy and thus propagate preferentially *perpendicular* to neuronal pathways. The WETCOW theory provides a comprehensive framework for characterizing the propagation of EM fields through the complex brain tissue microstructure and larger-scale morphology (e.g. cortical folding) and provides the dynamic solution to the electric potential field necessary to solve the EEG inverse problem.

The problem of spatially localizing the EEG signal involves estimating the most probable distribution of electric field amplitudes given an array of sensors. This is essentially a problem of correctly modeling the physics of how EM waves propagate through the complex environment of the convoluted brain tissue morphology and the anisotropic and inhomogeneous nature of brain tissue. The current state-of-the-art approach to this problem, called 'source localization', such as *low-resolution electromagnetic tomography* or LORETA algorithm with its many variations (*Pascual-Marqui et al., 1994*; *Hallez et al., 2005*; *Hallez et al., 2007*; *Dattola et al., 2020*), also called 'EEG source imaging' (*Michel et al., 2004*; *Ojeda et al., 2021*), involves using a predefined brain atlas, arbitrarily placing dipole sources on the surface, and calculating the contribution from these sources. Some methods propose using fMRI as a prior, which has the disadvantage of requiring fMRI acquisitions (*Rosa et al., 2010*; *Schultze-Kraft et al., 2011*; *Jorge et al., 2014*; *Abreu et al., 2022*). The current source localization methods are based on a static model for the electric field caused by a fixed set of predefined dipole sources (see *Michel and Brunet, 2019*, for a review of current methods). This model is inherently

limited because in reality the brain's electrical field variations are time dependent and generated by an essentially continuous distribution of sources through the entire brain. This description is the essence of the WETCOW theory (*Galinsky and Frank, 2020b*; *Galinsky and Frank, 2021*), which describes how highly coherent localized electric field phenomena, such as cortical wave loops and synchronized spiking, are produced by the complex nonlinear interactions of waves across multiple spatial and temporal scales.

The theoretical discussion above in Maxwell's equations in the brain clarifies that the problem being solved by SPECTRE is the inversion of a dynamical wave equation model in contrast to the solution of Poisson's equation for static sources being solved by traditional 'source localization' methods. Therefore, it makes little sense to compare the two methods, as they are attempting to solve two completely different problems. However, it perhaps serves some purpose to illustrate with a specific practical example a most basic computational problem encountered if one tries to even attempt a 'source localization' solution with the basis state even remotely resembling the setup used in SPECTRE.

In a typical application of the SPECTRE method, we use an MNI volumetric grid with 2 mm (902,629 voxels), 1 mm (7,221,032 voxels), or 0.73 mm (11,393,280 voxels). All voxels in our models are considered sites of EM activity consistent with the local intravoxel tissue characteristics (via *Equation 13*) rather than any assumed dipolar form used in source localization methods. This setup is facilitated by the pseudo-spectral computational approach used in SPECTRE detailed above in Numerical implementation. Our highest resolution (0.73 mm) SPECTRE processing can be completed on a modern workstation using 16–20 Gb of memory in a matter of minutes.

For comparison with a current state-of-the-art source localization method, we downloaded the currently available LoretaKey1 software (*Dattola et al., 2020*), which uses as a default a set of 6239 fixed dipoles. In order to make a fair comparison, we tried to use LoretaKey1 with a number of dipoles comparable to the number of voxels we use for our lower resolution ($2mm$ voxel) reconstruction in this paper. We began with the equivalent 902,629 voxels used by SPECTRE in the $2mm$ reconstruction, but found that LoretaKey1 is not able to handle this size because of memory limitations (i.e. 'out-of-memory' crashes). We subsequently scaled down the number of dipoles by factors of 2, 4, 10, and 20 times. Only with around 45K dipoles were we able to make Loreta run. It ran for approximately 24 hr, but then again crashed due to out-of-memory problems. And it should be kept in mind that this is in the absence of even any attempt by Loreta to provide a solution to a time-dependent wave equation. On the contrary, our processing with 2 mm requires around 650 Mb of memory and takes only a matter of minutes to complete. At this point, it was decided that it was not possible for Loreta to provide a result that would usefully inform the efficacy of the SPECTRE method.

## Discussion
### Neuroimaging and brain activity models

The ultimate goal of functional neuroimaging is to noninvasively detect and quantify the spatial and temporal variations in brain activity in terms of functional modes or 'networks'. This requires the development of models for brain activity for the quantities being measured by the imaging modality and a reconstruction method to estimate the parameters of the physical model from that modality's data.

The two methodologies that have emerged as the modalities of choice, fMRI and EEG, offer an interesting perspective on this general problem. The recognized importance of spatially localizing brain activity led to the development of fMRI in order to leverage the ability of MRI to spatially localize anatomical regions of the brain. The price paid is that the brain activity measured was constrained to be related to the physical effects that MRI was sensitive to, which turned out to be the local magnetic field perturbations produced by the susceptibility variations due to the changes in the oxygenation state of hemoglobin (*Ogawa et al., 1990*). The actual spatiotemporal effects measured in an fMRI experiment are a complicated combination of this effect filtered through the simultaneous influences of metabolic, blood flow, and biomechanical factors (*Buxton and Frank, 1997*). The connection of the fMRI to brain electrical activity is therefore quite indirect.

The problem faced by EEG is, in some sense, the opposite of fMRI. It directly measures the electrical activity of the brain, but does so using only measurements made by an array of electrodes placed on the surface of the head. There are no direct spatial localization capabilities with EEG. It is important to clarify what is meant by that statement. The ability of MRI to spatially localize signals from the brain

is based on a physical model of how the signal is related to the location. In essence, this boils down to the Larmor expression $\omega(x) = \gamma B(x)$, whose very simplicity belies the extraordinary history of quantum mechanics. MRI leverages this expression with an equally impressive history of engineering physics and computational science to produce modern scanners and sophisticated acquisition and analysis methods for reconstructing volumetric data. With this viewpoint, the limitations of EEG can be seen as the absence of an appropriate physical model for the generation and propagation of brain electrical signals on which to base a reconstruction, or 'inversion', to produce images. Previous models have invoked a 'quasi-static approximation' (*Nunez et al., 1997*) that precludes the existence of a more realistic dynamical brain wave model and limits what information can be extracted from EEG data. Our recently developed WETCOW model of brain electrodynamics derived from first principles revealed the existence of measurable brain waves that can permeate throughout the entire brain volume, not just along neuronal pathways. The model depends on the detailed morphology and tissue composition of an individual's brain. The SPECTRE method then leveraged this theory to develop a general method for reconstructing the modes of spatiotemporal brain electrical activity using a variety of additional estimation tools we have developed and along with tissue and morphology information provided by high-resolution MRI anatomical data. The result is a practical, numerically efficient, subject-specific method for directly reconstructing the time-dependent electrical activity throughout the entire brain volume directly from EEG measurements acquired by standard extant EEG systems.

## Comparison with fMRI

fMRI has become the de facto neuroimaging method for spatial and temporal localization of brain activity. The contrast mechanism that forms the basis of fMRI is the blood oxygenation level-dependent (BOLD) variations in the magnetic state of hemoglobin and its influence on the local MRI signal as a function of the local metabolism and hemodynamics (*Buxton and Frank, 1997*). Consequently, the spatial and temporal characteristics of the fMRI signal are related to blood flow and metabolic dynamics, rather than direct measures of electrical activity. In particular, the signal variations will be spatially localized in vascular pathways, and the temporal variations, being related to blood flow effects, are very slow compared to electrical activity. In short, the spatial-temporal dynamics measured by fMRI need not (and, in fact, will not) correspond exactly to the spatial-temporal patterns of electrical activity. Numerous experimental realities also make fMRI problematic as a gold standard. In particular, fMRI is facilitated by enhancing the sensitivity of MRI to the BOLD contrast mechanism, which requires enhancing the sensitivity to local magnetic field variations through the use of $T_2$-weighted pulse sequences (*Jezzard, 2012*), which lead to increased geometric distortions, compromising not only spatial resolution but confounding the spatial localization of the activity in a complex, nonlinear fashion. Gross distortions can lead to significantly reduced signal-to-noise (SNR) and even completely unrecoverable signal loss, particularly in regions near air/tissue interfaces, such as in the PFC. Moreover, the complex nonlinear interactions between the magnetic fields and physiological variations such as respiration and cardiac pulsations produce a variety of complex spatiotemporal signal distortions (*Frank et al., 2001*). While mitigating these artifacts is an area of very active research, they remain a serious problem for fMRI.

Nevertheless, certain very simple task-based fMRI experimental paradigms, such as finger tapping or rapidly flickering checkerboard stimuli, repeated at periodic on/off intervals, have been established as experiments that produce repeatable robust activations in known brain networks and are commonly used as basic testbeds for assessment of analysis algorithms. When combined with simultaneous EEG acquisition, such experiments provide two different types of data that can be compared as a form of validation, with the proviso that these two methods are imaging different physical quantities.

While the advantages of SPECTRE over fMRI in temporal resolution are clear, what is perhaps surprising is its advantages in *spatial* resolution. The inverse solution that estimates the electric field potential from the EEG data is based on a physical model of wave propagation from tissues whose composition and geometry are derived from high-resolution anatomical MRI data. The final resolution of the SPECTRE electric field modes is that of the anatomical data, which is typically significantly higher ($\sim .5 - 1mm$) than the resolution of an fMRI image ($\sim 2mm$). (There are, of course, limitations depending on the number of electrodes in the EEG system.)

But it is also important to recognize that the question of resolution in fMRI is not just a question of the prescribed image resolution of the acquisition. The BOLD physical mechanism that generates

the fMRI contrast is a subtle variation in the magnetic susceptibility, which causes variations in the local magnetic field, that in turn alters the local signal. fMRI acquisitions are specifically designed to accentuate this effect in order to make it observable. Unfortunately, local magnetic field variations unrelated to the BOLD mechanism, in particular strong magnetic susceptibility variations due to air/tissue boundaries such as those in the sinus cavities, cause severe nonlinear image distortions that effectively alter the location and shape of the affected image volume elements (voxels). This makes even the definition of 'resolution' problematic, as it is essentially a spatially nonlinearly varying function. Such effects are absent from EEG, which is simply a set of receiving electrodes (albeit not without its own source of artifacts) (*Nunez et al., 1997*). The SPECTRE reconstruction uses high-resolution MRI data acquired with techniques specifically designed to be insensitive to these magnetic susceptibility distortions and thus of very high spatial fidelity.

## Advantages of SPECTRE and future work

The SPECTRE reconstruction of EEG data provides obvious significant advantages over fMRI in temporal resolution, since EEG data has very high intrinsic temporal resolution ($\sim 1ms$) necessary to capture rapidly varying electric field variations. Moreover, the SPECTRE algorithm can specify what frequency ranges to interrogate, providing a highly flexible analysis framework for focused investigation of particular frequency bands of interest. On the contrary, even rapid fMRI acquisition is intrinsically limited by the temporal evolution of the contrast mechanism, the BOLD signal, which is related to blood flow and thus of quite low frequency ($\sim 1Hz$).

In this paper, we have successfully validated the SPECTRE method using simultaneous fMRI/EEG experiments. The results not only affirmed SPECTRE's capability to accurately reconstruct spatial distributions of neural activity from EEG data, in alignment with the concurrently acquired fMRI data, but also revealed its efficacy in identifying robust activations across subjects that were not detectable with fMRI alone. These findings underscore SPECTRE's potential to significantly enhance the sensitivity and scope of neuroimaging analyses. Further validation was performed using iEEG measurements from an epilepsy study, with reconstruction of data from a subset of sensors on the surface of the brain shown to be consistent with the reconstruction from all the sensors, including those directly next to the activity source. The application of SPECTRE to high-resolution EEG data during a gambling task demonstrated its ability to reconstruct a well-known and important brain circuit (*Olds and Milner, 1954*; *Leshner, 1997*; *Koob and Volkow, 2010*; *Gardner, 2011*; *Schultz, 2015*; *Haber, 2017*; *Banich and Floresco, 2019*) that has previously only been detected using fMRI. The analysis revealed significant differences in the brain networks in the alpha range $8 - 12Hz$, consistent with previous spatially resolved fMRI experiments, but the analysis is easily carried out in any user-defined frequency ranges of interest (*Hagerty et al., 2013*; *Cantisani et al., 2016*; *Stavropoulos and Carver, 2018*; *Leung and Pang, 2021*), which will be the subject of future work. The SPECTRE methodology is applicable to any EEG study and thus holds promise for a wide range of ongoing studies of basic neuroscience of reward mechanisms and in clinical applications such as addiction.

## Conclusion

The implications for spatially resolved EEG are important not only from a scientific perspective, but from a practical perspective as well. fMRI is a much more involved and expensive procedure, requiring highly trained research or clinical applications specialists in specially designed facilities, and subjecting the subjects to a much more claustrophobic and restricted environment, with the safety concerns always present in MRI experiments. On the contrary, the portability, safety, and relative ease of EEG experiments, which can be carried out in a standard research or clinical office, makes it very attractive. The high spatial and temporal resolution capabilities provided by SPECTRE to standard EEG data offer the possibility of more detailed investigations of brain activity in a wide range of both basic research and clinical settings. This method also has important implications for the democratization of medicine worldwide, where there are many populations for which advanced technologies such as fMRI are prohibitive because of cost, citing issues for large specialized equipment, and lack of highly trained personnel.

## Human subjects

All participants provided informed consent approved by the University of Victoria's Human Research Ethics Board. The iEEG data was recorded in drug-resistant epilepsy patients undergoing invasive EEG monitoring at the North Shore University Hospital (Manhasset, NY 11030, USA) for seizure onset localization. All patients provided informed written consent according to a protocol approved by the Institutional Review Board (IRB) of the Feinstein Institutes for Medical Research in accordance with the Declaration of Helsinki. All participants in the simultaneous fMRI/EEG study provided informed consent approved by the IRB of the Nathan Kline Institute for Psychiatric Research (Orangeburg, NY, USA).

## Code availability

The code supporting the findings of this study is protected by patent and university intellectual property regulations and, therefore, is not publicly available. Interested parties may contact Lawrence Frank (lfrank@ucsd.edu) to inquire about potential licensing options through UCSD.

## Acknowledgements

LRF and VLG were supported by NSF grants ACI-1550405 and AGS-2114860 and NIH grants R01-AG054049 and R01-AG079280. LRF was also supported by Simons Foundation grant AR-HU-MAN-00004264. ST was supported by NIH U24 AA021695, NIH U01 AA021692, NIH U01 DA041089, NIH R01 DA057567, and OK was supported by NSERC Discovery Grant RGPIN 2016-0943. AM was supported by (NIMH) R21MH123875. SB was supported by NIDCD R01DC019979. The technology in this paper is covered under US Patents 10909414, 10789713, and 11270445.

## Additional information

### Funding

| Funder | Grant reference number | Author |
| --- | --- | --- |
| National Institutes of Health | R01- AG054049 | Lawrence R Frank Vitaly L Galinsky |
| National Institutes of Health | R01-AG079280 | Lawrence R Frank Vitaly L Galinsky |
| National Science Foundation | ACI-1550405 | Lawrence R Frank Vitaly L Galinsky |
| National Science Foundation | AGS-2114860 | Lawrence R Frank Vitaly L Galinsky |
| Simons Foundation Autism Research Initiative | AR-HUMAN- 00004264 | Lawrence R Frank |
| National Institutes of Health | U24 AA021695 | Susan Tapert |
| National Institutes of Health | U01 AA021692 | Susan Tapert |
| National Institutes of Health | U01 DA041089 | Susan Tapert |
| National Institutes of Health | R01 DA057567 | Susan Tapert |
| Natural Sciences and Engineering Research Council of Canada | Discovery Grant RGPIN 2016-0943 | Olave Krigolson |
| National Institute on Deafness and Other Communication Disorders | R01DC019979 | Stephan Bickel |

| Funder | Grant reference number | Author |
| --- | --- | --- |
| National Institutes of Health | R21MH123875 | Antigona Martinez |

The funders had no role in study design, data collection and interpretation, or the decision to submit the work for publication.

## Author contributions

Lawrence R Frank, Conceptualization, Resources, Software, Formal analysis, Supervision, Funding acquisition, Validation, Investigation, Visualization, Methodology, Writing – original draft, Project administration, Writing – review and editing; Vitaly L Galinsky, Conceptualization, Resources, Data curation, Software, Formal analysis, Supervision, Funding acquisition, Validation, Investigation, Visualization, Methodology, Writing – original draft, Project administration, Writing – review and editing; Olave Krigolson, Stephan Bickel, Resources, Data curation, Validation, Investigation, Methodology, Writing – original draft, Writing – review and editing; Susan Tapert, Investigation, Methodology, Writing – original draft, Writing – review and editing; Antigona Martinez, Conceptualization, Resources, Data curation, Formal analysis, Funding acquisition, Validation, Investigation, Visualization, Methodology, Writing – original draft, Writing – review and editing

## Author ORCIDs
Lawrence R Frank ⓘ https://orcid.org/0000-0001-7235-587X
Vitaly L Galinsky ⓘ https://orcid.org/0000-0003-0420-6834

## Ethics

Human subjects: All participants provided informed consent approved by the University of Victoria's Human Research Ethics Board (UVic Approval Number: 16-428). The iEEG data was recorded in drug-resistant epilepsy patients undergoing invasive EEG monitoring at the North Shore University Hospital (Manhasset, NY 11030, USA) for seizure onset localization. All patients provided informed written consent according to a protocol approved by the IRB of the Feinstein Institutes for Medical Research in accordance with the Declaration of Helsinki. All participants in the simultaneous fMRI/EEG study provided informed consent approved by the Institutional Review Board (IRBNet ID #: 1698186) of the Nathan Kline Institute for Psychiatric Research (Orangeburg, NY).

Reviewer #1 (Public review): https://doi.org/10.7554/eLife.100123.3.sa1
Reviewer #2 (Public review): https://doi.org/10.7554/eLife.100123.3.sa2
Author response https://doi.org/10.7554/eLife.100123.3.sa3

# Additional files

## Supplementary files
MDAR checklist

## Data availability

The simultaneous EEG/fMRI data of Figures 1 and 2 (*Telesford et al., 2023*) is publicly available at https://fcon_1000.projects.nitrc.org/indi/retro/nat_view.html. The reward EEG data (*Williams et al., 2021*) of Figures 11,12, and 13 is publicly available at https://osf.io/65x4v/. For additional access to the reward data, contact Dr. Olave Krigolson (krigolson@uvic.ca). Please submit a proposal. The data cannot be used for commercial use, otherwise there are no restrictions on its use. Commercial research on the data in not allowed because this is prohibited by ethics approval. All participants that participated in the simultaneous fMRI/EEG study (Figures 3,4,5) provided informed consent approved by the Institutional Review Board (IRB) of the Nathan Kline Institute for Psychiatric Research (Orangeburg, NY). All participants that participated in the iEEG study (Figure 8, Appendix 2—figures 1–3) provided informed consent approved by the IRB of the Feinstein Institutes. Due to the sensitive nature of neuroimaging data collected from individuals with schizophrenia and iEEG data in patients with epilepsy, and to protect participant confidentiality and comply with IRB guidelines and HIPAA regulations, these data are not publicly available. De-identified data will be made available after reasonable

request to Antigona Martinez (martinez@nki.rfmh.org) for the fMRI/EEG data and to Stephan Bickel (sbickel@northwell.edu) for the iEEG data. Access upon reasonable request will be granted without restriction on researcher affiliation or location, but use of the data for commercial purposes is not permitted. The code supporting the findings of this study is protected by patent and university intellectual property regulations and, therefore, is not publicly available. Interested parties may contact Lawrence Frank (lfrank@ucsd.edu) to inquire about potential licensing options through UCSD.

The following previously published datasets were used:

| Author(s) | Year | Dataset title | Dataset URL | Database and Identifier |
|---|---|---|---|---|
| Williams CC, Ferguson TD, Hassall CD, Abimbola W, Krigolson OE | 2021 | The ERP, Frequency, and Time-Frequency Correlates of Feedback Processing: Insights from a Large Sample Study | https://osf.io/65x4v/ | Open Science Framework, 65x4v |
| Telesford QK, Gonzalez-Moreira E, Xu T, Tian Y, Colcombe SJ, Cloud J, Russ BE, Falchier A, Nentwich M, Mad- sen J, Parra LC, Schroeder CE, Milham MP, Franco AR | 2023 | EEG/FMRI Naturalistic Viewing Dataset | https://fcon_1000. projects.nitrc.org/ indi/retro/nat_view. html | FCP/INDI, nat_view |

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

# Appendix 1

## Entropy field decomposition

This appendix is reproduced with permission under the terms of the Creative Commons Attribution license (CC BY 4.0) from *Frank et al., 2024*.

### Inverse problems in dynamical systems

Analysis of brain activity can be considered from two viewpoints. One view is the theoretical construction of the equations governing brain dynamics that when implemented computationally facilitates numerical exploration of the effects of different neurophysiological parameters on the final brain states. This is the *forward problem*. The other view is attempting to reconstruct the actual brain activity that was observed with real measured data. This is the *inverse problem*. These viewpoints are complementary, as the comparison of measurements with predicted results from theoretical models is the basis for the refinement of theoretical models of the brain, while the incorporation of more refined models into the reconstruction problem can produce more accurate quantitative assessments of actual brain states.

Solving inverse problems from data acquired from measurements of nonlinear systems is a well-known challenge in many, if not most, scientific disciplines where real data are collected. Neuroimaging data is particularly challenging because it varies over so many spatial and temporal scales and has so many interacting parameters (e.g. electric and magnetic fields, local metabolism, blood flow at arterial, vascular, and capillary scales, etc.). This is in addition to the usual challenges of inverse problems, such as finite sampling and both instrument and environmental noise. Indeed, the analysis of dynamic nonlinear systems poses a very general analysis problem across multiple scientific disciplines.

### Probabilistic approach

Characterizing the complex dynamics of a physical system by estimating the parameters of a hypothesized physical model falls under the purview of probability theory and was the motivation for our development of the EFD, described in detail in *Frank and Galinsky, 2016a*; *Frank and Galinsky, 2016b*. Here, we present a more intuitive overview (based on *Frank and Galinsky, 2016a*) and limit the discussion to details pertinent to neuroimaging data.

The neuroimaging data we are interested in are four-dimensional, sampled at three spatial dimensions and one time dimension. Denoting the spatial sampling as $\{n_x, n_y, n_z\}$, the total number of points in a volume is $N = n_x \times n_y \times n_z$. Measurements from this volume are made at a set of discrete times $t_i, i = 1, \ldots, n$ (assumed to be at equal intervals, though this is not a requirement of the analysis.) The data $\{d_{i,j}\}, i = 1, \ldots, n, \; j = 1, \ldots, N$ can equivalently be represented as $\{d_l\}$, where $\xi_l, l = 1, \ldots nN$ defines a set of space-time locations. Each data point is of the form

$$d_{i,j} = R s_{i,j} + e_{i,j} \tag{14}$$

where $R$ is an operator that represents the response of the measurement system to a signal $s_{i,j}$, and $e_{i,j}$ is the noise with the covariance matrix $\Sigma_e = \langle ee^\dagger \rangle$, where $\dagger$ means the complex conjugate transpose.

In NWP, the model equations are run forward in time using a set of initial conditions to generate the signal $s$, from which can be constructed data $d$ by finite sampling the generated signal and adding noise and instrument constraints. For the inverse (or estimation) problem, the goal is to instead determine the signal $s$ from the actual measured data $d$. This procedure is codified using Bayes' theorem to construct the posterior probability of the signal $s$, given the data $d$, and any information $I$ that is known about the problem, including any assumptions or hypotheses $H$:

$$\underbrace{p(s|d, I, H)}_{Posterior} = \frac{\overbrace{p(d|s, I, H)}^{Likelihood} \overbrace{p(s|I, H)}^{Prior}}{\underbrace{P(d|I, H)}_{Evidence}} \tag{15}$$

The most probable signal is the one reconstructed from the configuration of estimated parameters to produce the maximum in this posterior distribution. The underlying signal is assumed

to be continuous in space-time, and thus characterized by a field $\psi(x, t) \equiv \psi(\xi)$ although the data consist of discrete samples in both space and time so that the reconstructed underlying signal is $s_l = \int \psi(\xi)\delta(\xi - \xi_l)d\xi$.

While the Bayesian approach to signal analysis is not new (for an excellent introduction, see **Bretthorst, 1988**), there have remained practical issues that have limited its applicability to nonlinear interacting fields. By interacting, we are referring to fields that influence one another. For instance, in fMRI, blood flow at multiple scales (e.g. vascular and capillary) and electrophysiology are coupled and influence each other. The limitations in practical applications can be thought of as falling into two broad categories: (1) How to represent nonlinear interacting fields, and (2) How to incorporate relevant prior information. One approach to the first problem is reformulating Bayes' theorem in terms of field theory (**Enßlin et al., 2009**), called *information field theory* (IFT), which facilitates the use of techniques developed in the physics discipline of field theory (e.g. **Ryder, 1985**). The advantage, and disadvantage, of this method is that it is essentially flexible in its ability to approximate any desired number of parameters and field interactions. This reformulation can be done by rewriting Bayes' theorem (**Equation 15**) in the form (**Enßlin et al., 2009**)

$$p(\psi|d, I) = \frac{e^{-H(d,\psi)}}{Z(d)} \tag{16}$$

where the field theoretic quantities, the Hamiltonian $H(d, \psi)$ and the partition function $Z(d)$, are

$$H(d, \psi) = -\ln p(d, \psi|I) \tag{17a}$$

$$Z(d) = p(d|I) = \int d\psi\, e^{-H(d,\psi)} \tag{17b}$$

The partition function is a *generating function* from which can be constructed expressions for all orders of field interactions. For the current purposes, however, it will be considered a constant that can be ignored. The Hamiltonian describes the conserved quantities in field theories, and in IFT describes the conservation of probability. This will be the central quantity of interest. It is written

$$H(d, \psi) = H_0 - j^\dagger \psi + \frac{1}{2}\psi^\dagger D^{-1} \psi + H_i(d, \psi) \tag{18}$$

where $H_0$ is essentially a normalizing constant that can be ignored, $D$ is an information propagator, $j$ is an information source, and $H_i$ is an interaction term (Equation 7 in **Frank and Galinsky, 2016a**). The solution for the fields is the minimum of the Hamiltonian ($\delta H/\delta\psi = 0$), which is

$$\psi(\xi) = D\left(j - H_i'\right) \tag{19}$$

where $H_i'$ symbolically represents the variation of $H_i$ (the second term on the right-hand side of Equation 8 in **Frank and Galinsky, 2016a**).

The expressions, **Equations 18; 19**, are not in a form clearly equivalent to the more standard formulations of probability theory, so it is useful to demonstrate their equivalence in a simple problem. This will then make apparent where it deviates from standard formulations for use in more complex problems.

Consider the simple case in which the signal is assumed to be described by a set of model functions $F$, with each component weighted by an amplitude $a$. If the functions $F$ do not interact with one another or the noise, the interaction term $H_i'$ in **Equation 19** can be ignored. Ignoring instrument effect ($R = 1$) and assuming the signal is contaminated by zero mean Gaussian noise with variance $\Sigma_e = \sigma^2$, the solution is (**Frank and Galinsky, 2016a**)

$$\psi(\xi) = Dj \quad \text{where} \quad \begin{cases} D = \sigma^2(F^\dagger F)^{-1} \\ j = \sigma^{-2}F^\dagger d \end{cases} \tag{20}$$

The source $j$ is noise-weighted projection of the signal onto the sampled model functions (sometimes called the 'dirty map' [**Tan, 1986**]), and the propagator $D$ characterizes the influence of

the noise and the sampling of the functions $F$ (sometimes called the 'dirty beam' [*Tan, 1986*]). The estimated signal is

$$\psi(\xi) = F^+ d \quad \text{where} \quad F^+ = (F^\dagger F)^{-1} F^\dagger \qquad (21)$$

where $F^+$ is the well-known pseudo-inverse. *Equation 21* is just the standard maximum a posterior result (*Jaynes, 2003*). The rationale for the names 'source' and 'propagator' becomes clear. The input data $d(t_i)$ projected along the $k$'th component of the model function $F(t_i)$ provides the source $j$ of new information, which is then propagated by $D$ from which estimates of the field components are derived.

If the physical system were describable in terms of noninteracting plane waves, the $F$ would be the standard Fourier functions, then the source is just the noise-weighted inverse Fourier transform of the data, while the propagator is the covariance of the sampled Fourier model functions. Each Fourier component would constitute a 'mode' of the system, which would be a four-dimensional (three spatial and one temporal) time-varying volume. These modes would be ranked according to their amplitude, or eigenvalue. This result is called the 'Fourier decomposition' of the data and is ubiquitous across a wide range of scientific disciplines. It is important to note that this Bayesian analysis reveals that the estimate of the spatiotemporal patterns is *not* just the inverse Fourier transform of the data, but takes into account the uncertainties related to the sampling and the noise via the propagator $D$.

Unfortunately, many, perhaps most, physical systems are not easily characterized by such simple models. This is certainly the case for the brain electrodynamics considered in this paper, where a more appropriate general model is that of nonlinear, nonperiodic, interacting fields. This makes the process of estimating the system from the data much more difficult, because many more combinations of the model parameters are consistent with the data. In addition, nonlinear systems can be exquisitely sensitive to initial conditions, which are rarely known precisely and again can produce a wide range of results for the same set of system parameters. The IFT provides a framework to formally characterize complex interacting nonlinear fields, but no guidance on resolving these ambiguities. The resulting field theoretic description produces expressions with an essentially infinite number of terms. Without a method for effectively ranking the relative importance of the terms, problems can quickly become intractable.

## The role of prior information

Solving seemingly intractable inverse problems requires the reduction in the number of possible system configurations (i.e. the set of parameters characterizing the system) that are consistent with the data. This can be achieved through the development of increasingly refined theoretical physical models. But probability theory provides another powerful mechanism, which is the ability to incorporate prior information. Formally, this is done through the 'prior' in *Equation 15*. However, the apparent simplicity of this term belies the complexity in using it. One common and conceptually simple use is to incorporate statistical parameters from previous measurements that characterize a distribution known, or estimated, to describe the system. For example, for a system whose parameters are assumed to be normally distributed, estimates of average and standard deviation can be used to construct a Gaussian prior distribution. While such an approach is common, it is not particularly useful when systems are highly nonlinear and non-Gaussian.

One of the remarkable traits of the brain's electrical activity is that, despite its seemingly often 'random' behavior (e.g. neuronal avalanches), it displays the remarkable ability to produce highly synchronized behavior in space and time that characterize coherent modes of activity. Characterizing such coherences is one of the major objectives of neuroimaging data analysis. Observations of such data bring up a subtle but important issue that arises in the estimation of coherent space-time processes. By coherence, we mean that there are spatial patterns that are persistent in time. For example, a tornado is a persistent spatially localized vorticity. While these notions are intuitively clear, they are notoriously difficult to formalize in analytical theories of estimation. Consider, for example, the idealized numerical simulation shown in *Appendix 1—figure 1* in which there are two areas of activity in Gaussian random noise: a point oscillating with very high SNR (centered on the red dot) but also a larger circular region with very low SNR (centered at the blue dot). The computation question can be phrased in terms of 'What is the significance of the activated regions?' In more intuitive terms, while the high SNR region clearly seems to indicate an area of activity, we

intuitively understand that the larger region is significant despite its low amplitude because it is so persistently coherent over such a large spatial region. Is there a way to incorporate the spatial coherence into the computation of significance?

## Entropy spectrum pathways

The problem then is this: How does one identify regions of spatial and temporal coherence in a dataset that contains complex spatiotemporal patterns that are unique from event to event and thus not amenable to fitting by some standard model? This was the problem faced by Lorenz, who recognized that the implications of the unpredictability of atmospheric systems for estimation theory required a general method that could be applied to any dataset from such systems, which led him to the formulation of *empirical orthogonal functions* (*Lorenz, 1956*), which is synonymous with the PCA (*Pearson, 1901*). On the face of it, PCA is 'model free' in that the physical model is not parameterized. However, it is based on the assumption that the data can be described by a multivariate Gaussian distribution and proceeds by successively fitting the data to $n$ ellipsoids following the subtraction of the previous $n-1$ components. While PCA is easy to implement, it typically is a poor model for complex physical systems, and the resulting components therefore can often have little relationship to the actual coherent modes within the data. Therefore, it makes identification of physically relevant parameters more problematic.

Remarkably, one answer to this question of how to incorporate prior information on system coherences is hidden in plain sight - it is contained within the data itself. Spatial and temporal coherences are characterized by being significantly correlated with their neighbors. In an image, this could simply be the similarity of intensities in neighboring spatial locations. In neuroimaging data, it might take the form of similar waveform vectors in neighboring regions of both space and time. This concept can be formalized by computing space-time correlations from the data, which are local interactions (i.e. computed in adjacent spatial and temporal points in the data) but, when computed at every location in a dataset, also provide information about the larger-scale coherent structures in the data. The relationship between these local interactions and the large-scale structures in the data is the subject of the theory of ESPs (*Frank and Galinsky, 2014*).

The essence of the theory is that the coupling between adjacent (in space or time) data elements $i$ and $j$ can be viewed as a measure of the information that can be transmitted between those two locations. One could imagine computing the correlation between the copper content in adjacent elements of a volumetric image of a transmission cable. The resulting high probability regions of adjacent high correlations would be the copper wire, and therefore literally be the path of maximum information transmission. This concept can be formalized and abstracted to any dataset using a parameter or set of parameters. The local interactions reveal the large-scale pathways in the data and therefore the pathways of information transmission about the coherences of these parameters. This concept is formalized by computing a *coupling matrix* $Q_{ij}$ between adjacent data elements $i$ and $j$ and computing its eigenvectors $\phi^{(k)}$, $k = 1, \ldots, M$. (For a detailed discussion of the construction of $Q_{ij}$, see *Frank and Galinsky, 2016a*.) Remarkably, these eigenvectors can be interpreted as the pathways between locations that can be reached in the most number of ways. In other words, the path in the data that have maximum path entropy. In the wire example, an electron starting at any location on the wire would preferentially travel along that wire. The eigenvector with the largest eigenvalue ($k = 1$), the principal eigenvector, defines the maximum entropy pathway, that which is most preferred.

A simple example of ESP to uncover spatial correlations is shown in *Appendix 1—figure 2* (top), where a simple 2D spatial curve is detected with the principal eigenvector of the coupling matrix formed from the intensities at each location. This is to be compared with the standard PCA result in *Appendix 1—figure 2* (middle) that required multiple components to be computed and still only approximately detects the correct structure. The extension to space-time is shown in *Appendix 1—figure 2* (bottom), where the principal ESP eigenvector correctly picks out the preferred space-time pathway. PCA in this case (not shown) becomes even more problematic.

The recognition that non-Gaussian systems are poorly described by PCA leads to the formulation of a nonlinear version called ICA, which seeks to decompose data into independent non-Gaussian components. However, this 'model-free' method also contains hidden assumptions that produce unreliable results in nonlinear complex physical systems (see *Frank and Galinsky, 2016b*, for a more extended discussion and examples).

## The entropy field decomposition

The ESP prior can be incorporated into the estimation scheme by using the coupling matrix $Q_{ij}$ as the prior in *Equation 15*

$$p(s|d, I) = \frac{1}{2\pi Q^{1/2}} \exp\left(-\frac{1}{2} s_i^\dagger Q_{ij} s_j\right) \tag{22}$$

which then becomes part of the Hamiltonian in *Equation 16*. (For a detailed discussion of the construction of $Q_{ij}$, see *Frank and Galinsky, 2016a*.) The result is that the significant spatial and temporal correlations within the data can provide highly relevant prior information about the most probable parameter configurations of a physical system and select out only the limited number of relevant terms within the infinite number of choices that make the inverse problem consistent with the data. The incorporation of the ESP theory into the Bayesian framework of IFT is called the EFD.

Returning to the simple example above, with the ESP prior, the solution given in *Equation 20* is the same but with a modified propagator that now includes the coupling matrix:

$$D = \left[Q + F^\dagger \Sigma_e^{-1} F\right]^{-1} \tag{23}$$

This means that the estimated modes of the system are now dependent on the coupling matrix $Q$. And in particular, the optimal solutions depend on the eigenvectors $\phi$ of $Q$. Therefore, the EFD modes are found by expressing the probability, i.e., the Hamiltonian, in terms of $\phi$. In other words, whereas in the simplest, uncoupled problem above, the Fourier functions serve as the proper 'basis' to describe the problem, now the eigenfunction of the $Q$ serves that function. These eigenvectors incorporate both the spatial and the temporal correlations, and thus address the problem shown in *Appendix 1—figure 1*: the significance of the detected activated region is enhanced by the adjacency, or clustering, of the activated voxels, as we intuitively believe. Therefore, no post hoc clustering of voxels determined independently to be activated based on their time course is required, which is the current widespread methodology for detecting activation in, for example, fMRI data (*Nichols and Holmes, 2002*; *Winkler et al., 2014*; *Eklund et al., 2016*; *Wang et al., 2021*).

This extends to the more general solution where interactions between fields are considered as well, and the interaction Hamiltonian $H_i'$ (*Equation 19* is not ignored see *Frank and Galinsky, 2016a*, for details). In general, solving the eigenvalue problem for $Q$ produces a *transition probability* $p_{ijk}$ between locations $(i, j)$ for the $k$'th mode and an *equilibrium distribution* $\mu^{(k)}$ associated with the set of $k = 1, M$ mode eigenvectors $\phi^{(k)}$. The EFD procedure produces $M$ modes that are ranked by their eigenvalues, providing a simple, unsupervised method for characterizing the primary modes of data collected from an arbitrarily complex nonlinear, nonperiodic, non-Gaussian physical system.

A compelling feature of the EFD approach is that the basis functions *are unique to every problem*, since they are derived directly from the data. This makes the EFD approach powerful for physical systems that are complex nonlinear systems, such as brain neural activity. Moreover, the key practical issue is that the coupling matrix $Q$ can be defined by the user according to the problem at hand. The details of incorporating multiple parameters into the coupling matrix are discussed in *Galinsky and Frank, 2017*. A schematic of the EFD procedure is shown in *Appendix 1—figure 3*.

## Multi-modality EFD (JESTER)

Extending the EFD methods to multiple modalities by incorporating coupling between different parameters, which we call JESTER (*Galinsky and Frank, 2017*), is accomplished as follows. For $m = 1, ..., M$, different modalities $d^{(m)}$ with the coupling matrices $Q^{(m)}$ that all correspond to the same unknown signal $s$, intermodality coupling matrix can be constructed as the product of the coupling matrices for the individual modalities expressed in the ESP basis and registered to a common reference frame, which we denote $\widetilde{Q}^{(m)}$: i.e., the joint coupling matrix is $\mathcal{Q}^{(m)} = \prod_m \widetilde{Q}^{(m)}$. More specifically, the joint coupling matrix $\mathcal{Q}_{ij}$ between any two space-time locations $(i, j)$ can be written in the general (equivalent) form as

$$\ln \mathcal{Q}_{ij} = \sum_{m=1}^{M} \beta_{ij}^{(m)} \ln \widetilde{Q}_{ij}^{(m)} \tag{24}$$

where the exponents $\beta^{(m)}$ can either be some constants or functions of data collected for different modalities $\beta_{ij}^{(m)} \equiv \beta^{(m)}(\widetilde{\boldsymbol{d}}_i, \widetilde{\boldsymbol{d}}_j), \widetilde{\boldsymbol{d}}_i \equiv \{\widetilde{d}_i^{(1)}, ..., \widetilde{d}_i^{(M)}\}$, where $\widetilde{d}_i^{(m)}$ and $\widetilde{Q}_{ij}^{(m)}$ represent, respectively, the data and the coupling matrix of the modality dataset $m$ represented in the ESP basis and evaluated at locations $r_i$ and $r_j$ of a common reference domain $R$:

$$\widetilde{d}_i^{(m)} = d^{(m)}\left(\psi^{(m)}(r_i)\right), \qquad \widetilde{Q}_{ij}^{(m)} = Q^{(m)}\left(\psi^{(m)}(r_i), \psi^{(m)}(r_j)\right) \tag{25}$$

where $\psi^{(m)} : R \rightarrow X$ denotes a diffeomorphic mapping of $m$th modality from the reference domain $R$ to an acquisition space $X$. In the general EEG reconstruction problem, the coupling of modalities can include the EEG data, the high-resolution anatomical MRI data, fMRI data, and dMRI data (*Galinsky et al., 2018*).

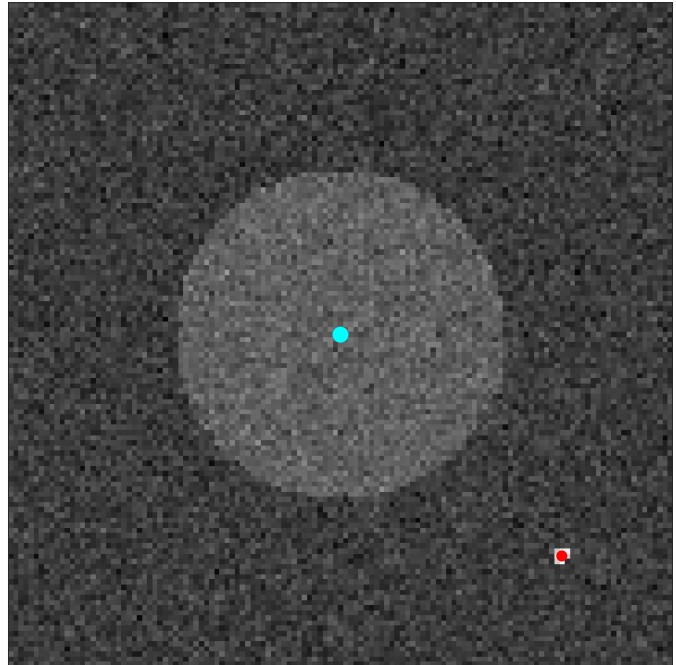

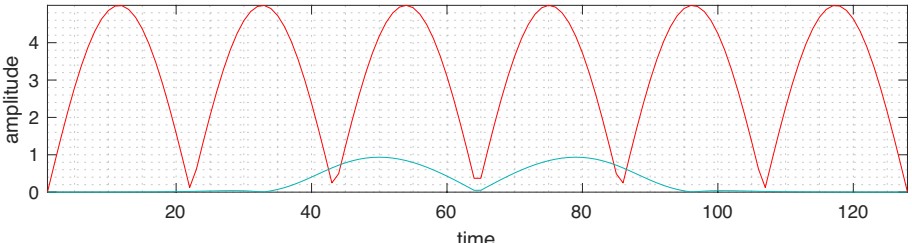

**Appendix 1—figure 1.** The significance of a detected spatiotemporal pattern. Idealized numerical simulation in which there are two areas of activity in Gaussian random noise: a point oscillating with very high signal-to-noise (SNR) (centered on the red dot) but also a larger circular region with very low SNR (centered at the blue dot). Traditional estimation methods tend to favor high SNR signals (red dot) but have difficulty with low SNR activity with very high spatial correlations (blue dot). The entropy field decomposition (EFD) takes both spatial and temporal correlations into account and therefore detects both regions.

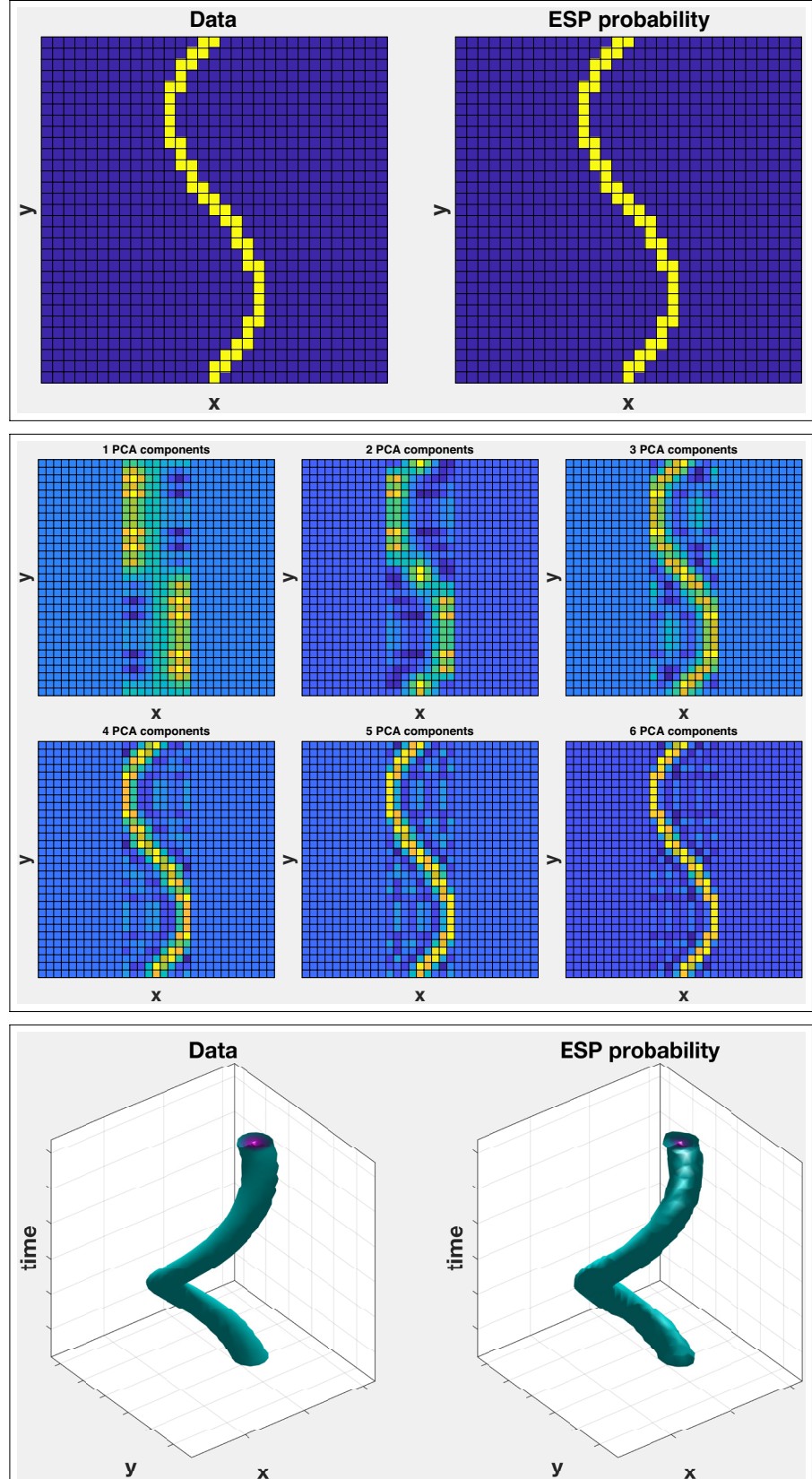

**Appendix 1—figure 2.** (Top left) Original lattice; (top right) entropy spectrum pathway (ESP) probability. (Middle) Principal components analysis (PCA) results on same original lattice. The ESP probability locates the structure in *Appendix 1—figure 2 continued on next page*

a single calculation. The PCA decomposition, even for six components, shows significant errors. Several more components would be required to accurately fit the data. (Bottom row) Space-time ESP. (Top left) Space-time trajectory of 2D data; (top right) ESP probability.

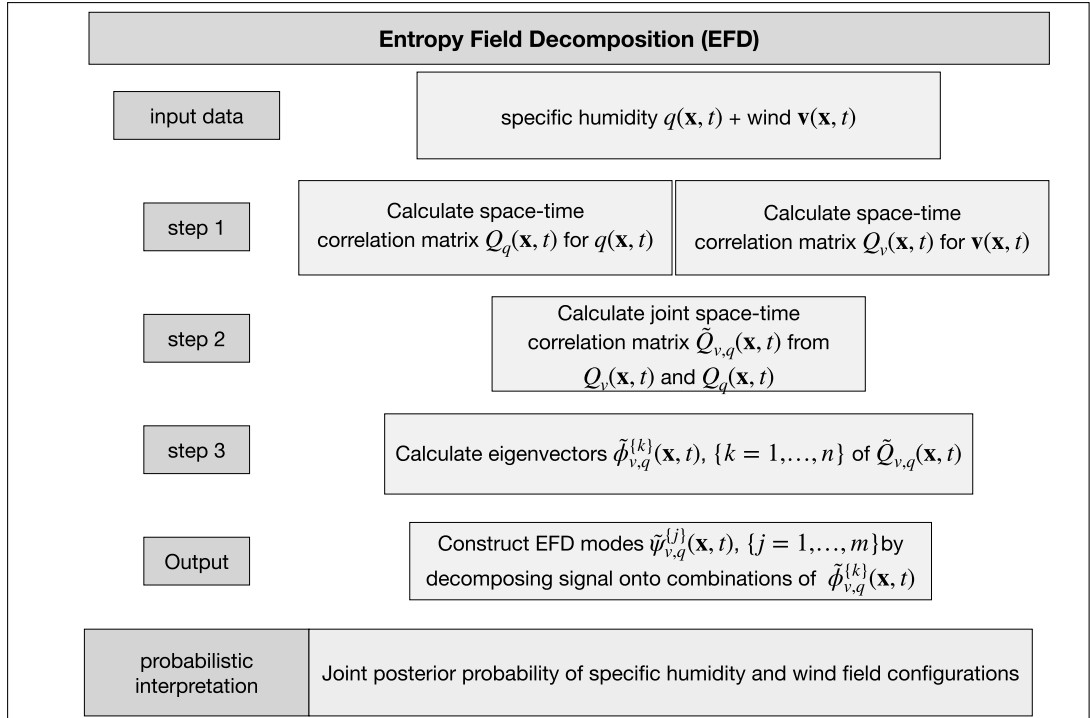

**Appendix 1—figure 3.** Schematic for the construction of space-time entropy field decomposition (EFD) modes from the data.

# Appendix 2

## Data and methods

### Data

#### Visual paradigm data

For complete details of the fMRI and EEG acquisitions, see *Telesford et al., 2023*. Minimal details for the EEG and MRI acquisitions are provided here for continuity.

#### EEG acquisitions

Duplicated from *Telesford et al., 2023*. EEG is collected using a customized cap to record 61 cortical channels, two electrooculogram (EOG) channels placed above (channel 64) and below the left eye (channel 63), and one electrocardiography (ECG) channel (channel 32) placed on the back. In addition, the cap also contains a reference and ground electrode. Electrodes were filled using V19 Abralyt HiCl electrode gel. Electrode impedance was kept below 20 kOhm. EEG was recorded using BrainVision Recorder at a sampling rate of 5 kHz.

#### MRI acquisitions

Duplicated from *Telesford et al., 2023*. MRI data were acquired using a 12-channel head coil on 3.0T Siemens TIM Trio. MPRAGE structural T1w images were acquired with the following parameters: TR = 2500 ms; TI = 1200 ms; TE = 2.5 ms; slices = 192; matrix size = 256 × 256; voxel size = 1 mm$^3$ isotropic; flip angle = 8°; partial Fourier off; pixel bandwidth = 190 Hz/Px. All BOLD fMRI sequences were acquired with these parameters: TR = 2100 ms; TE = 24.6 ms; flip angle = 60°; slices = 38; matrix size = 64 × 64; voxel size = 3.469 × 3.469 × 3.330 mm$^3$.

#### Attention paradigm data

#### Simultaneous EEG/fMRI acquisition

fMRI and structural MRI images were acquired on a Siemens 3T TIM-Trio scanner (NKI Center for Biomedical Imaging and Neuromodulation) equipped with a 32-channel phased array head coil. Structural T1 and T2 scans were collected using standard sequences. Whole-brain BOLD data was acquired with a gradient-echo EPI sequence (TR = 2000 ms; TE = 30 ms; flip angle = 80°). EEG data were acquired concurrently with fMRI using an MR-compatible EEG amplifier (BrainVision MR series, Brain Products, Munich, Germany) and a 64-channel MR-compatible ring electrode cap with 10–20 International System electrode placement cap. EEG data was sampled at a rate of 5 kHz EEG data were acquired at a rate of 5 kHz using BrainVision Recorder software (Brain Products). Electrocardiographic data were captured from electrodes on the backs of subjects. The reference electrode was positioned between Fz and Cz. Scanner and heartbeat artifacts were removed offline from the EEG signal using an average template subtraction procedure (*Niazy et al., 2005*), and the data was resampled to 250 Hz.

#### Traditional EEG analysis

The single-trial EEG signal from each electrode was convolved with a 3-cycle Morlet wavelet computed over a 3 s window centered at the onset of each stimulus and averaged separately for each stimulus type. The averaged spectral amplitude at each time point was then baseline-corrected by subtracting the mean spectral amplitude over the −200 to −50 prestimulus interval. Further details of postprocessing and time-frequency analyses methods are described in *Lakatos et al., 2005*; *Martínez et al., 2015*; *Martínez et al., 2019*.

#### Reward circuit data

#### Task and design acquisition

Participants completed a simple gambling task (*Williams et al., 2021*). On each trial, they saw a black fixation cross for 500 ms, followed by two colored squares for 500 ms, and then the fixation cross turned gray (go cue), and participants were to select one of the two squares (square locations - left, right - were randomized on each trial) within a 2000 ms time limit. They were then presented with a black fixation cross for 300–500 ms, and then, simple feedback as to their performance ('WIN' for gain, 'LOSE' for loss) for 1000 ms in black font. If the participants responded before the go cue, they were instead delivered 'TOO FAST' feedback, and if they did not respond before the 2000 ms time limit, it would be considered a loss. The goal of the participants was to accumulate wins by

determining which of the two squares would more often lead to gains (60% vs 10%). In this task, participants accumulated wins; however, were not paid money. They would see the same pair of colors for one block of 20 trials. They conducted six blocks of unique color pairs.

## Participants

Five hundred undergraduate students were included and were recruited via the University of Victoria psychology participant pool (see *Williams et al., 2021*, for details). The data was collected until 500 participants became available that were not characterized by one of the following a priori criteria: trial count after artifact rejection was less than 15 per condition, total artifact rejection exceeded 40% of trials rejected, FCz (electrode of interest) specific artifact rejection exceeded 40% of trials rejected, or independent component analysis-based blink correction failed. These criteria were extremely strict to ensure clean data in the analyses, and as such, a total of 637 participants were analyzed before reaching the goal of 500 clean participants. All participants had normal or corrected-to-normal vision and volunteered to take part in the experiment for extra course credit in a psychology course. All participants provided informed consent approved by the University of Victoria's Human Research Ethics Board.

## Data acquisition and preprocessing

Data were re-referenced to an average mastoid reference and filtered using a 0.1–30 Hz passband (Butterworth, order 4) and a 60 Hz notch filter. Correction for eye blinks was performed using EEGLAB's ICA. Components reflective of blinks were manually identified and removed via topographic maps and component loadings, and data were reconstructed. Data were then segmented from $-500$ to 1500 ms relative to feedback stimulus onset, baseline-corrected using a $-200$ to $0 ms$ window, and run through artifact rejection with $10 \mu V/ms$ gradient and $100 \mu V$ maximum-minimum criteria. Data were preprocessed to identify noisy or damaged electrodes using artifact rejection trial removal rates for each electrode.

The 1 s of recorded sequence for each 'WIN' or 'LOSE' event were extracted from recordings for each participant (with 22 ms of pre-event sample and 488 ms of post-event sample) and combined together to form separate winning and losing datasets. Each of those datasets was processed using SPECTRE to construct the approximate inverse solution for the potential $\phi$ across an entire 2 mm MNI brain volume.

### iEEG validation: additional subjects
The analysis presented in *Appendix 2—figures 1–3* is the same analysis for Subjects 2–4 as presented for Subject 1 in *Figure 8*.

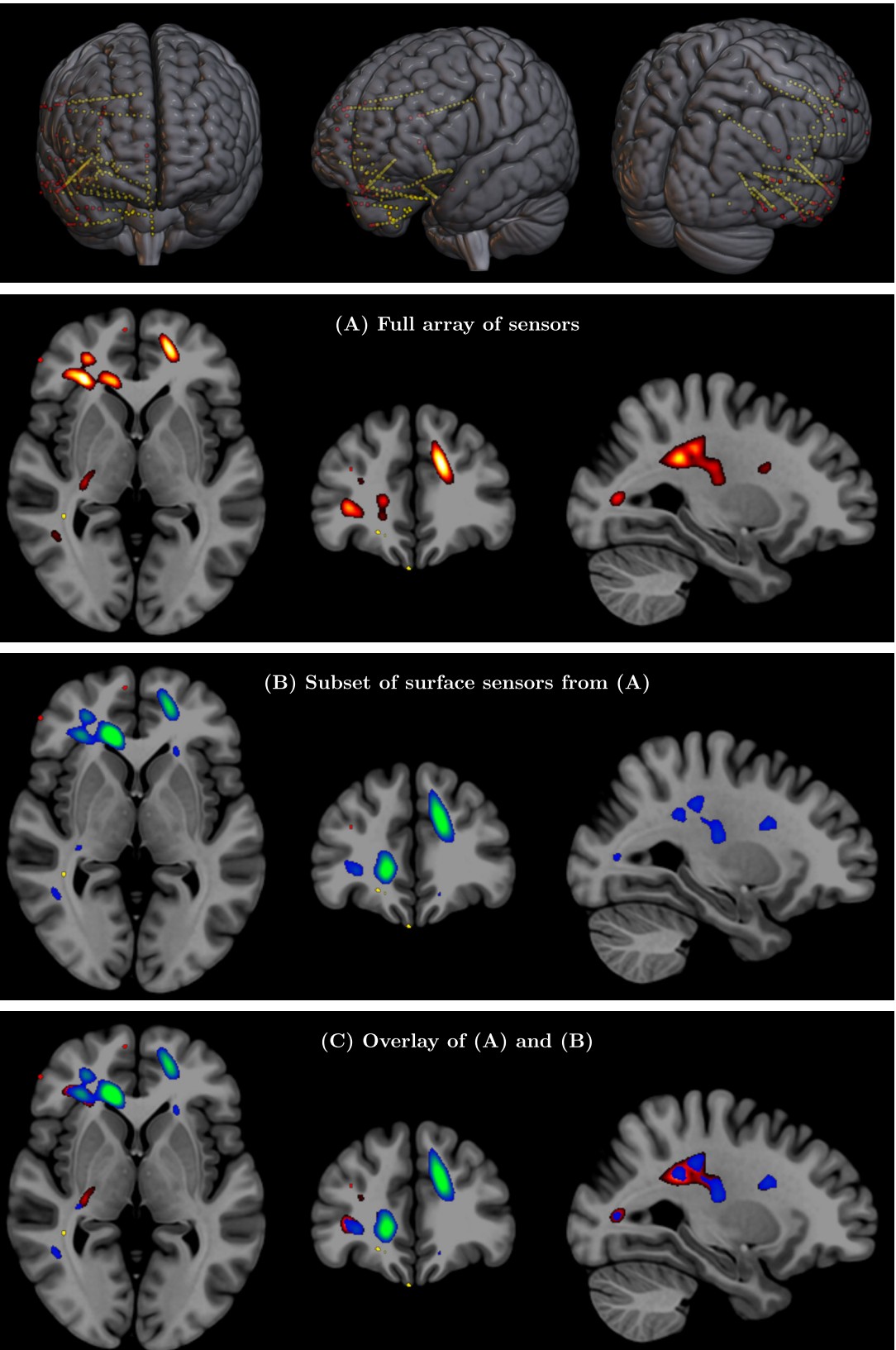

**Appendix 2—figure 1.** Validation of SPatially resolved EEG Constrained with Tissue properties by Regularized Entropy (SPECTRE) with intracranial electroencephalography (iEEG) data for Subject 2 (see *Figure 8* for details).

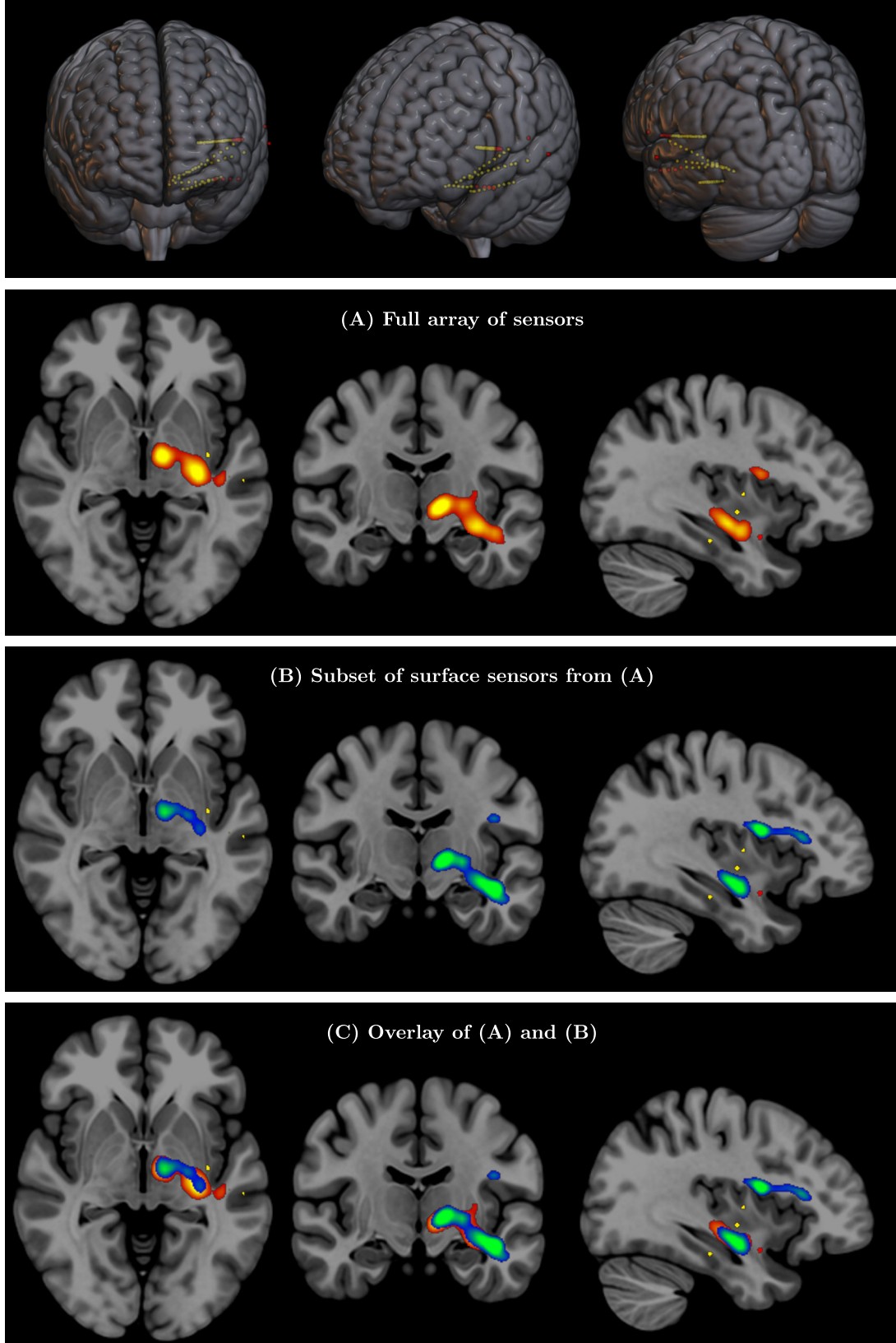

**Appendix 2—figure 2.** Validation of SPatially resolved EEG Constrained with Tissue properties by Regularized Entropy (SPECTRE) with intracranial electroencephalography (iEEG) data for Subject 3 (see *Figure 8* for details).

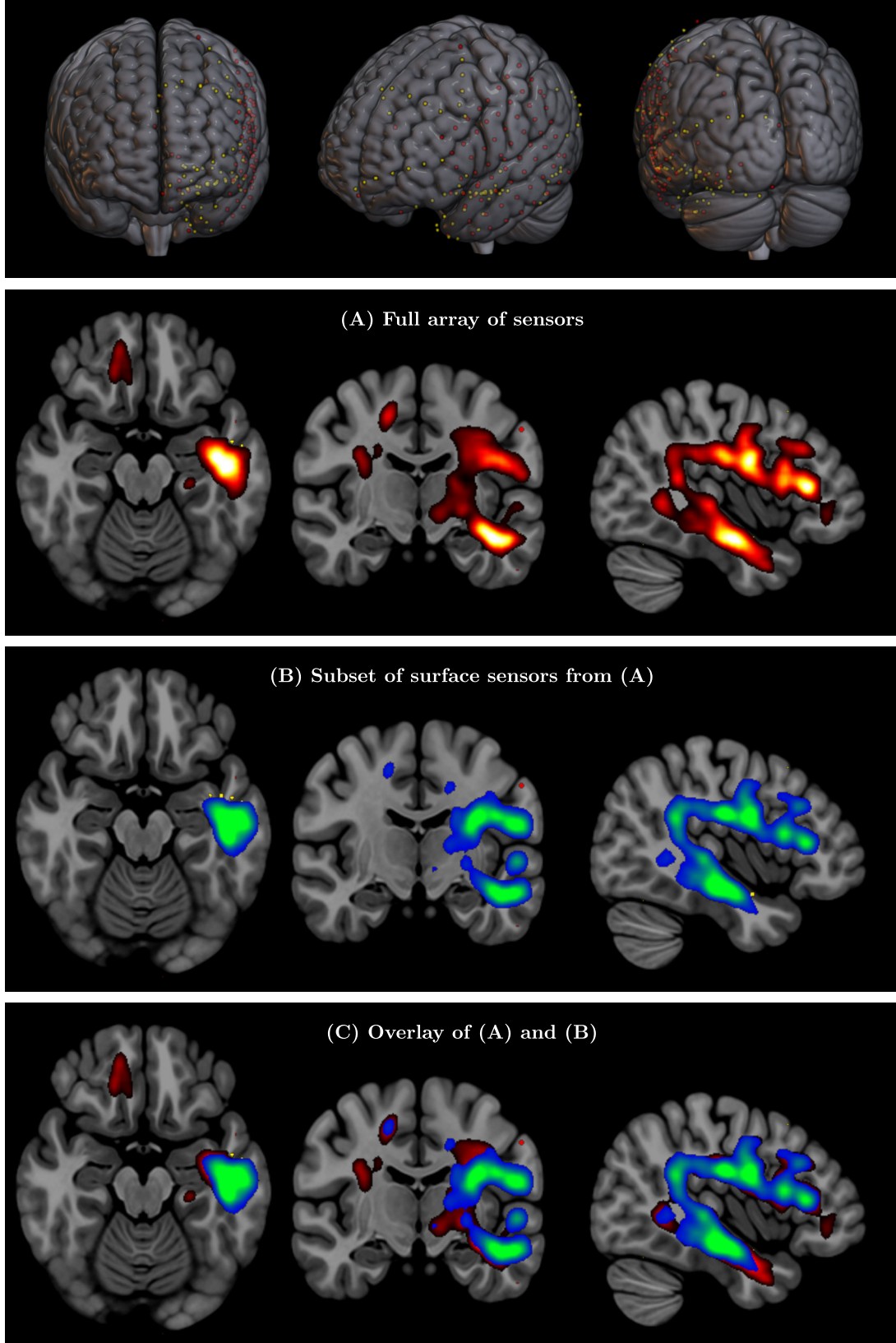

**Appendix 2—figure 3.** Validation of SPatially resolved EEG Constrained with Tissue properties by Regularized Entropy (SPECTRE) with intracranial electroencephalography (iEEG) data for Subject 4 (see *Figure 8* for details).

