## [Editor Report · eLife Assessment]

This **fundamental** work has the potential to advance our understanding of brain activity using electrophysiological data, by proposing a completely new approach to reconstructing EEG data that challenges the assumptions typically made in the solutions to Maxwell’s equations. **Convincing** evidence for the superior spatio-temporal resolution of this method is provided through a number of experiments, including simultaneous FMRI/EEG acquisitions. This work will be of broad interest to neuroscientists and neuroimaging.

---

## [Referee Report · Reviewer #1 (Public review)]

I want to reiterate my comment from the first round of reviews: that I am insufficiently familiar with the intricacies of Maxwell's equations to assess the validity of the assumptions and the equations being used by WETCOW. The work ideally needs assessing by someone more versed in that area, especially given the potential impact of this method if valid.

Effort has been made in these revisions to improve explanations of the proposed approach (a lot of new text has been added) and to add new simulations.

However, the authors have still not compared their method on real data with existing standard approaches for reconstructing data from sensor to physical space. Refusing to do so because existing approaches are deemed inappropriate (i.e. they "are solving a different problem") is illogical.

Similarly, refusing to compare their method with existing standard approaches for spatio-temporally describing brain activity, just because existing approaches are deemed inappropriate, is illogical.

For example, the authors say that "it's not even clear what one would compare [between the new method and standard approaches]". How about:

(1) Qualitatively: compare EEG activation maps. I.e. compare what you would report to a researcher about the brain activity found in a standard experimental task dataset (e.g. their gambling task). People simply want to be able to judge, at least qualitatively on the same data, what the most equivalent output would be from the two approaches. Note, both approaches do not need to be done at the same spatial resolution if there are constraints on this for the comparison to be useful.

and

(2) Quantitatively: compare the correlation scores between EEG activation maps and fMRI activation maps

The abstract claims that there is a "direct comparison with standard state-of-the-art EEG analysis in a well-established attention paradigm", but no actual comparison appears to have been completed in the paper.

---

## [Referee Report · Reviewer #2 (Public review)]

Summary:

The manuscript claims to present a novel method for direct imaging of electric field networks from EEG data with higher spatiotemporal resolution than even fMRI. Validation of the EEG reconstructions with EEG/FMRI, EEG, and iEEG datasets are presented. Subsequently, reconstructions from a large EEG datasets of subjects performing a gambling task are presented.

Strengths:

If true and convincing, the proposed theoretical framework and reconstruction algorithm can revolutionise the use of EEG source reconstructions.

Weaknesses:

There is very little actual information in the paper about either the forward model or the novel method of reconstruction. Only citations to prior work by the authors are given with absolutely no benchmark comparisons, making the manuscript difficult to read and interpret in isolation to their prior body of work.

Comments on revisions:

This is a major rewrite of the paper. The authors have improved the discourse vastly. There is now a lot of didactics included but they are not always relevant to the paper. The section on Maxwell's equation does a disservice to the literature in prior work in bioelectromagnetism and does not even address the issues raised in classic text books by Plonsey et al. There is no logical "backwardness" in the literature. They are based on the relative values of constants in biological tissues. Several sections of the appendix discuss in terms of weather predictions and could just be written specifically for the problem here. There are reinventions of many standard ideas in terms of physics discourses, like Bayesian theory or PCA etc. I think that the paper remains quite opaque and many of the original criticisms remain, especially as they relate to multimodal datasets. The overall algorithm still remains poorly described. The comparisons to benchmark remain unaddressed and the authors state that they couldn't get Loreta to work and so aborted that. The figures are largely unaltered, although they have added a few more, and do not clearly depict the ideas. Again, no benchmark comparisons are provided to evaluate the results and the performance in comparison to other benchmarks.

---

## [Author Response]

The following is the authors’ response to the current reviews.

**Reviewer 1 (Public Review):**
I want to reiterate my comment from the first round of reviews: that I am insufficiently familiar with the intricacies of Maxwell’s equations to assess the validity of the assumptions and the equations being used by WETCOW. The work ideally needs assessing by someone more versed in that area, especially given the potential impact of this method if valid.

We appreciate the reviewer’s candor. Unfortunately, familiarity with Maxwell’s equations is an essential prerequisite for assessing the veracity of our approach and our claims.

Effort has been made in these revisions to improve explanations of the proposed approach (a lot of new text has been added) and to add new simulations. However, the authors have still not compared their method on real data with existing standard approaches for reconstructing data from sensor to physical space. Refusing to do so because existing approaches are deemed inappropriate (i.e. they “are solving a different problem”) is illogical.

Without understanding the importance of our model for brain wave activity (cited in the paper) derived from Maxwell’s equations in inhomogeneous and anisotropic brain tissue, it is not possible to critically evaluate the fundamental difference between our method and the standard so-called “source localization” method which the Reviewer feels it is important to compare our results with. Our method is not “source localization” which is a class of techniques based on an inappropriate model for static brain activity (static dipoles sprinkled sparsely in user-defined areas of interest). Just because a method is “standard” does not make it correct. Rather, we are reconstructing a whole brain, time dependent electric field potential based upon a model for brain wave activity derived from first principles. It is comparing two methods that are “solving different problems” that is, by definition, illogical.

Similarly, refusing to compare their method with existing standard approaches for spatio-temporally describing brain activity, just because existing approaches are deemed inappropriate, is illogical.

Contrary to the Reviewer’s assertion, we do compare our results with three existing methods for describing spatiotemporal variations of brain activity.

First, Figures 1, 2, and 6 compare the spatiotemporal variations in brain activity between our method and fMRI, the recognized standard for spatiotemporal localization of brain activity. The statistical comparison in Fig 3 is a quantitative demonstration of the similarity of the activation patterns. It is important to note that these data are simultaneous EEG/fMRI in order to eliminate a variety of potential confounds related to differences in experimental conditions.

Second, Fig 4 (A-D) compares our method with the most reasonable “standard” spatiotemporal localization method for EEG: mapping of fields in the outer cortical regions of the brain detected at the surface electrodes to the surface of the skull. The consistency of both the location and sign of the activity changes detected by both methods in a “standard” attention paradigm is clearly evident. Further confirmation is provided by comparison of our results with simultaneous EEG/fMRI spatial reconstructions (E-F) where the consistency of our reconstructions between subjects is shown in Fig 5.

Third, measurements from intra-cranial electrodes, the most direct method for validation, are compared with spatiotemporal estimates derived from surface electrodes and shown to be highly correlated.

For example, the authors say that “it’s not even clear what one would compare [between the new method and standard approaches]”. How about:(1) Qualitatively: compare EEG activation maps. I.e. compare what you would report to a researcher about the brain activity found in a standard experimental task dataset (e.g. their gambling task). People simply want to be able to judge, at least qualitatively on the same data, what the most equivalent output would be from the two approaches. Note, both approaches do not need to be done at the same spatial resolution if there are constraints on this for the comparison to be useful.(2) Quantitatively: compare the correlation scores between EEG activation maps and fMRI activation maps

These comparison were performed and already in the paper.

(1) Fig 4 compares the results with a standard attention paradigm (data and interpretation from Co-author Dr Martinez, who is an expert in both EEG and attention). Additionally, Fig 12 shows detected regions of increased activity in a well-known brain circuit from an experimental task (’reward’) with data provided by Co-author Dr Krigolson, an expert in reward circuitry.

(2) Correlation scores between EEG and fMRI are shown in Fig 3.

(3) Very high correlation between the directly measured field from intra-cranial electrodes in an epilepsy patient and those estimated from only the surface electrodes is shown in Fig 9.

There are an awful lot of typos in the new text in the paper. I would expect a paper to have been proof read before submitting.

We have cleaned up the typos.

The abstract claims that there is a “direct comparison with standard state-of-the-art EEG analysis in a well-established attention paradigm”, but no actual comparison appears to have been completed in the paper.

On the contrary, as mentioned above, Fig 4 compares the results of our method with the state-of-the-art surface spatial mapping analysis, with the state-of-the-art time-frequency analysis, and with the state-of-the-art fMRI analysis

**Reviewer 2 (Public Review):**
This is a major rewrite of the paper. The authors have improved the discourse vastly.There is now a lot of didactics included but they are not always relevant to the paper.

The technique described in the paper does in fact leverage several novel methods we have developed over the years for analyzing multimodal space-time imaging data. Each of these techniques has been described in detail in separate publications cited in the current paper. However, the Reviewers’ criticisms stated that the methods were non-standard and they were unfamiliar with them. In lieu of the Reviewers’ reading the original publications, we added a significant amount of text indeed intended to be didactic. However, we can assume the Reviewer that nothing presented was irrelevant to the paper. We certainly had no desire to make the paper any longer than it needed to be.

The section on Maxwell’s equation does a disservice to the literature in prior work in bioelectromagnetism and does not even address the issues raised in classic text books by Plonsey et al. There is no logical “backwardness” in the literature. They are based on the relative values of constants in biological tissues.

This criticism highlights the crux of our paper. Contrary to the assertion that we have ignored the work of Plonsey, we have referenced it in the new additional text detailing how we have constructed Maxwell’s Equations appropriate for brain tissue, based on the model suggested by Plonsey that allows the magnetic field temporal variations to be ignored but not the time-dependence electric fields.

However, the assumption ubiquitous in the vast prior literature of bioelectricity in the brain that the electric field dynamics can be “based on the relative values of constants in biological tissues”, as the Reviewer correctly summarizes, is precisely the problem. Using relative average tissue properties does not take into account the tissue anisotropy necessary to properly account for correct expressions for the electric fields. As our prior publications have demonstrated in detail, taking into account the inhomogeneity and anisotropy of brain tissue in the solution to Maxwell’s Equations is necessary for properly characterizing brain electrical fields, and serves as the foundation of our brain wave theory. This led to the discovery of a new class of brain waves (weakly evanescent transverse cortical waves, WETCOW).

It is this brain wave model that is used to estimate the dynamic electric field potential from the measurements made by the EEG electrode array. The standard model that ignores these tissue details leads to the ubiquitous “quasi-static approximation” that leads to the conclusion that the EEG signal cannot be spatial reconstructed. It is indeed this critical gap in the existing literature that is the central new idea in the paper.

There are reinventions of many standard ideas in terms of physics discourses, like Bayesian theory or PCA etc.

The discussion of Bayesian theory and PCA is in response to the Reviewer complaint that they were unfamiliar with our entropy field decomposition (EFD) method and the request that we compare it with other “standard” methods. Again, we have published extensively on this method (as referenced in the manuscript) and therefore felt that extensive elaboration was unnecessary. Having been asked to provide such elaboration and then being pilloried for it therefore feels somewhat inappropriate in our view. This is particularly disappointing as the Reviewer claims we are presenting “standard” ideas when in fact the EFD is new general framework we developed to overcome the deficiencies in standard “statistical” and probabilistic data analysis methods that are insufficient for characterizing non-linear, nonperiodic, interacting fields that are the rule, rather than the exception, in complex dynamical systems, such as brain electric fields (or weather, or oceans, or ....).

The EFD is indeed a Bayesian framework, as this is the fundamental starting point for probability theory, but it is developed in a unique and more general fashion than previous data analysis methods. (Again, this is detailed in several references in the papers bibliography. The Reviewer’s requested that an explanation be included in the present paper, however, so we did so). First, Bayes Theorem is expressed in terms of a field theory that allows an arbitrary number of field orders and coupling terms. This generality comes with a penalty, which is that it’s unclear how to assess the significance of the essentially infinite number of terms. The second feature is the introduction of a method by which to determine the significant number of terms automatically from the data itself, via the our theory of entropy spectrum pathways (ESP), which is also detailed in a cited publication, and which produces ranked spatiotemporal modes from the data. Rather than being “reinventions of many standard ideas” these are novel theoretical and computational methods that are central to the EEG reconstruction method presented in the paper.

I think that the paper remains quite opaque and many of the original criticisms remain, especially as they relate to multimodal datasets. The overall algorithm still remains poorly described. benchmarks.

It’s not clear how to assess the criticisms that the algorithm is poorly described yet there is too much detail provided that is mistakenly assessed as “standard”. Certainly the central wave equations that are estimated from the data are precisely described, so it’s not clear exactly what the Reviewer is referring to.

The comparisons to benchmark remain unaddressed and the authors state that they couldn’t get Loreta to work and so aborted that. The figures are largely unaltered, although they have added a few more, and do not clearly depict the ideas. Again, no benchmark comparisons are provided to evaluate the results and the performance in comparison to other benchmarks.

As we have tried to emphasize in the paper, and in the Response to Reviewers, the standard so-called “source localization” methods are NOT a benchmark, as they are solving an inappropriate model for brain activity. Once again, static dipole “sources” arbitrarily sprinkled on pre-defined regions of interest bear little resemblance to observed brain waves, nor to the dynamic electric field wave equations produced by our brain wave theory derived from a proper solution to Maxwell’s equations in the anisotropic and inhomogeneous complex morphology of the brain.

The comparison with Loreta was not abandoned because we couldn’t get it to work, but because we could not get it to run under conditions that were remotely similar to whole brain activity described by our theory, or, more importantly, by an rationale theory of dynamic brain activity that might reproduce the exceedingly complex electric field activity observed in numerous neuroscience experiments.

We take issue with the rather dismissive mention of “a few more” figures that “do not clearly depict the idea” when in fact the figures that have been added have demonstrated additional quantitative validation of the method.

The following is the authors’ response to the original reviews.

**Public Reviews:**

**Reviewer 1 (Public Review):**
The paper proposes a new source reconstruction method for electroencephalography (EEG) data and claims that it can provide far superior spatial resolution than existing approaches and also superior spatial resolution to fMRI. This primarily stems from abandoning the established quasi-static approximation to Maxwell’s equations.The proposed method brings together some very interesting ideas, and the potential impact is high. However, the work does not provide the evaluations expected when validating a new source reconstruction approach. I cannot judge the success or impact of the approach based on the current set of results. This is very important to rectify, especially given that the work is challenging some long- standing and fundamental assumptions made in the field.

We appreciate the Reviewer’s efforts in reviewing this paper and have included a significant amount of new text to address their concerns.

I also find that the clarity of the description of the methods, and how they link to what is shown in the main results hard to follow.

We have added significantly more detail on the methods, including more accessible explanations of the technical details, and schematic diagrams to visualize the key processing components.

I am insufficiently familiar with the intricacies of Maxwell’s equations to assess the validity of the assumptions and the equations being used by WETCOW. The work therefore needs assessing by someone more versed in that area. That said, how do we know that the new terms in Maxwell’s equations, i.e. the time-dependent terms that are normally missing from established quasi-static-based approaches, are large enough to need to be considered? Where is the evidence for this?

The fact that the time-dependent terms are large enough to be considered is essentially the entire focus of the original papers [7,8]. Time-dependent terms in Maxwell’s equations are generally not important for brain electrodynamics at physiological frequencies for homogeneous tissues, but this is not true for areas with stroung inhomogeneity and ansisotropy.

I have not come across EFD, and I am not sure many in the EEG field will have. To require the reader to appreciate the contributions of WETCOW only through the lens of the unfamiliar (and far from trivial) approach of EFD is frustrating. In particular, what impact do the assumptions of WETCOW make compared to the assumptions of EFD on the overall performance of SPECTRE?

We have added an entire new section in the Appendix that provides a very basic introduction to EFD and relates it to more commonly known methods, such as Fourier and Independent Components Analyses.

The paper needs to provide results showing the improvements obtained when WETCOW or EFD are combined with more established and familiar approaches. For example, EFD can be replaced by a first-order vector autoregressive (VAR) model, i.e. y_t_ = Ay_t−1_ + e_t_ (where y_t_ is [num_gridpoints_ ∗ 1] and A is [num_gridpoints_ ∗ num_gridpoints_] of autoregressive parameters).

The development of EFD, which is independent of WETCOW, stemmed from the necessity of developing a general method for the probabilistic analysis of finitely sampled non-linear interacting fields, which are ubiquitous in measurements of physical systems, of which functional neuroimaging data (fMRI, EEG) are excellent examples. Standard methods (such as VAR) are inadequate in such cases, as discussed in great detail in our EFD publications (e.g., [12,37]). The new appendix on EFD reviews these arguments. It does not make sense to compare EFD with methods which are inappropriate for the data.

The authors’ decision not to include any comparisons with established source reconstruction approaches does not make sense to me. They attempt to justify this by saying that the spatial resolution of LORETA would need to be very low compared to the resolution being used in SPECTRE, to avoid compute problems. But how does this stop them from using a spatial resolution typically used by the field that has no compute problems, and comparing with that? This would be very informative. There are also more computationally efficient methods than LORETA that are very popular, such as beamforming or minimum norm.

he primary reason for not comparing with ’source reconstruction’ (SR) methods is that we are are not doing source reconstruction. Our view of brain activity is that it involves continuous dynamical non-linear interacting fields througout the entire brain. Formulating EEG analysis in terms of reconstructing sources is, in our view, like asking ’what are the point sources of a sea of ocean waves’. It’s just not an appropriate physical model. A pre-chosen limited distribution of static dipoles is just a very bad model for brain activity, so much so that it’s not even clear what one would compare. Because in our view, as manifest in our computational implementation, one needs to have a very high density of computational locations throughout the entire brain, including white matter, and the reconstructed modes are waves whose extent can be across the entire brain. Our comments about the low resolution of computational methods for SR techniques really is expressing the more overarching concern that they are not capable of, or even designed for, detecting time-dependent fields of non-linear interacting waves that exist everywhere througout the brain. Moreover, the SR methods always give some answer, but in our view the initial conditions upon which those methods are based (pre-selected regions of activity with a pre-selected number of ’sources’) is a highly influential but artificial set of strong computational constraints that will almost always provide an answer consist with (i.e., biased toward) the expectations of the person formlating the problem, and is therefore potentially misleading.

In short, something like the following methods needs to be compared:(1) Full SPECTRE (EFD plus WETCOW)(2) WETCOW + VAR or standard (“simple regression”) techniques(3) Beamformer/min norm plus EFD(4) Beamformer/min norm plus VAR or standard (“simple regression”) techniques

The reason that no one has previously ever been able to solve the EEG inverse problem is due to the ubiquitous use of methods that are too ’simple’, i.e., are poor physical models of brain activity. We have spent a decade carefully elucidating the details of this statement in numerous highly technical and careful publications. It therefore serves no purpose to return to the use of these ’simple’ methods for comparison. We do agree, however, that a clearer overview of the advantages of our methods is warranted and have added significant additional text in this revision towards that purpose.

This would also allow for more illuminating and quantitative comparisons of the real data. For example, a metric of similarity between EEG maps and fMRI can be computed to compare the performance of these methods. At the moment, the fMRI-EEG analysis amounts to just showing fairly similar maps.

We disagree with this assessment. The correlation coefficient between the spatially localized activation maps is a conservative sufficient statistic for the measure of statistically significant similarity. These numbers were/are reported in the caption to Figure 5, and have now also been moved to, and highlighted in, the main text.

There are no results provided on simulated data. Simulations are needed to provide quantitative comparisons of the different methods, to show face validity, and to demonstrate unequivocally the new information that SPECTRE can ’potentially’ provide on real data compared to established methods. The paper ideally needs at least 3 types of simulations, where one thing is changed at a time, e.g.:(1) Data simulated using WETCOW plus EFD assumptions(2) Data simulated using WETCOW plus e.g. VAR assumptions(3) Data simulated using standard lead fields (based on the quasi-static Maxwell solutions) plus e.g. VAR assumptionsThese should be assessed with the multiple methods specified earlier. Crucially the assessment should be quantitative showing the ability to recover the ground truth over multiple realisations of realistic noise. This type of assessment of a new source reconstruction method is the expected standard

We have now provided results on simulated data, along with a discussion on what entails a meaningful simulation comparison. In short, our original paper on the WETCOW theory included a significant number of simulations of predicted results on several spatial and temporal scales. The most relevant simulation data to compare with the SPECTRE imaging results are the cortical wave loop predicted by WETCOW theory and demonstrated via numerical simulation in a realistic brain model derived from high resolution anatomical (HRA) MRI data. The most relevant data with which to compare these simulations are the SPECTRE recontruction from the data that provides the closest approximation to a “Gold Standard” - reconstructions from intra-cranial EEG (iEEG). We have now included results (new Fig 8) that demonstrate the ability of SPECTRE to reconstruct dynamically evolving cortical wave loops in iEEG data acquired in an epilepsy patient that match with the predicted loop predicted theoretically by WETCOW and demonstrated in realistic numerical simulations.

The suggested comparison with simple regression techniques serves no purpose, as stated above, since that class of analysis techniques was not designed for non-linear, non-Gaussian, coupled interacting fields predicted by the WETCOW model. The explication of this statement is provided in great detail in our publications on the EFD approach and in the new appendix material provided in this revision. The suggested simulation of the dipole (i.e., quasi-static) model of brain activity also serves no purpose, as our WETCOW papers have demonstrated in great detail that is is not a reasonable model for dynamic brain activity.

**Reviewer 2 (Public Review):**
Strengths:If true and convincing, the proposed theoretical framework and reconstruction algorithm can revolutionize the use of EEG source reconstructions.Weaknesses:There is very little actual information in the paper about either the forward model or the novel method of reconstruction. Only citations to prior work by the authors are cited with absolutely no benchmark comparisons, making the manuscript difficult to read and interpret in isolation from their prior body of work.

We have now added a significant amount of material detailing the forward model, our solution to the inverse problem, and the method of reconstruction, in order to remedy this deficit in the previous version of the paper.

**Recommendations for the authors:**

**Reviewer 1 (Recommendations):**
It is not at all clear from the main text (section 3.1) and the caption, what is being shown in the activity patterns in Figures 1 and 2. What frequency bands and time points etc? How are the values shown in the figures calculated from the equations in the methods?

We have added detailed information on the frequency bands reconstructed and the activity pattern generation and meaning. Additional information on the simultaneous EEG/fMRI acquisition details has been added to the Appendix.

How have the activity maps been thresholded? Where are the color bars in Figures 1 and 2?

We have now included that information in new versions of the figures. In addition, the quantitative comparison between fMRI and EEG are presented is now presented in a new Figure 2 (now Figure 3).

P30 “This term is ignored in the current paper”. Why is this term ignored, but other (time-dependent) terms are not?

These terms are ignored because they represent higher order terms that complicate the processing (and intepretation) but do not substatially change the main results. A note to this effect has been added to the text.

The concepts and equations in the EFD section are not very accessible (e.g. to someone unfamiliar with IFT).

We have added a lengthy general and more accessible description of the EFD method in the Appendix.

Variables in equation 1, and the following equation, are not always defined in a clear, accessible manner. What is i,δij,I,ω,F^ω,ϵ?

We have added additional information on how Eqn 1 (now Eqn 3) is derived, and the variables therein.

In the EFD section, what do you mean conceptually by α, i.e. “the coupled parameters α”?

This sentence has been eliminated, as it was superfluous and confusing.

How are the EFD and WETCOW sections linked mathematically? What is ψ (in eqn 2) linked to in the WETCOW section (presumably ϕ_ω_?) ?

We have added more introductory detail at the beginning of the Results to describe the WETCOW theory and how this is related to the inverse problem for EEG.

What is the difference between data d and signal s in section 6.1.3? How are they related?

We have added a much more detailed Appendix A where this (and other) details are provided.

What assumptions have been made to get the form for the information Hamiltonian in eqn3?

Eq 3 (now Eqn A.5) is actually very general. The approximations come in when constructing the interaction Hamiltonian H_i_.

P33 “using coupling between different spatio-temporal points that is available from the data itself” I do not understand what is meant by this.

This was a poorly worded sentence, but this section has now been replaced by Appendix A, which now contains the sentence that prior information “is contained within the data itself”. This refers to the fact that the prior information consists of correlations in the data, rather than some other measurements independent of the original data. This point is emphasized because in many Bayesian application, prior information consists of knowledge of some quantity that were acquired independently from the data at hand (e.g., mean values from previous experiments)

**Reviewer 2 (Recommendations):**
AbstractThe first part presents validation from simultaneous EEG/fMRI data, iEEG data, and comparisons with standard EEG analyses of an attention paradigm. Exactly what constitutes adequate validation or what metrics were used to assess performance is surprisingly absent.Subsequently, the manuscript examines a large cohort of subjects performing a gambling task and engaging in reward circuits. The claim is that this method offers an alternative to fMRI.IntroductionProvocative statements require strong backing and evidence. In the first paragraph, the “quasi-static” assumption which is dominant in the field of EEG and MEG imaging is questioned with some classic citations that support this assumption. Instead of delving into why exactly the assumption cannot be relaxed, the authors claim that because the assumption was proved with average tissue properties rather than exact, it is wrong. This does not make sense. Citations to the WETCOW papers are insufficient to question the quasi-static assumption.

The introduction purports to validate a novel theory and inverse modeling method but poorly outlines the exact foundations of both the theory (WETCOW) and the inverse modeling (SPECTRE) work.

We have added a new introductory subsection (“A physical theory of brain waves”) to the Results section that provides a brief overview of the foundations of the WETCOW theory and an explicit description of why the quasi-static approximation can be abandoned. We have expanded the subsequent subsection (“Solution to the inverse EEG problem”) to more clearly detail the inverse modeling (SPECTRE) method.

Section 3.2 Validation with fMRIFigure 1 supposedly is a validation of this promising novel theoretical approach that defies the existing body of literature in this field. Shockingly, a single subject data is shown in a qualitative manner with absolutely no quantitative comparison anywhere to be found in the manuscript. While there are similarities, there are also differences in reconstructions. What to make out of these discrepancies? Are there distortions that may occur with SPECTRE reconstructions? What are its tradeoffs? How does it deal with noise in the data?

It is certainly not the case that there are no quantitative comparisons. Correlation coefficients, which are the sufficient statistics for comparison of activation regions, are given in Figure 5 for very specific activation regions. Figure 9 (now Figure 11) shows a t-statistic demonstrating the very high significance of the comparison between multiple subjects. And we have now added a new Figure 7 demonstrating the strongly correlated estimates for full vs surface intra-cranial EEG reconstructions. To make this more clear, we have added a new section “Statistical Significance of the Results”.

We note that a discussion of the discrepancies between fMRI and EEG was already presented in the Supplementary Material. Therein we discuss the main point that fMRI and EEG are measuring different physical quantities and so should not be expected to be identical. We also highlight the fact that fMRI is prone to significant geometrical distortions for magnetic field inhomogeities, and to physiological noise. To provide more visibility for this important issue, we have moved this text into the Discussion section.

We do note that geometric distortions in fMRI data due to suboptimal acquisitions and corrections is all too common. This, coupled with the paucity of open source simultaneous fMRI-EEG data, made it difficult to find good data for comparison. The data on which we performed the quantitative statistical comparison between fMRI and EEG (Fig 5) was collected by co-author Dr Martinez, and was of the highest quality and therefore sufficient for comparison. The data used in Fig 1 and 2 was a well publicized open source dataset but had significant fMRI distortions that made quantitative comparison (i.e., correlation coefficents between subregions in the Harvard-Oxford atlas) suboptimal. Nevertheless, we wanted to demonstrate the method in more than one source, and feel that visual similarity is a reasonble measure for this data.

Section 3.2 Validation with fMRIFigure 2 Are the sample slices being shown? How to address discrepancies? How to assume that these are validations when there are such a level of discrepancies?

It’s not clear what “sample slices” means. The issue of discrepancies is addressed in the response to the previous query.

Section 3.2 Validation with fMRIFigure 3 Similar arguments can be made for Figure 3. Here too, a comparison with source localization benchmarks is warranted because many papers have examined similar attention data.

Regarding the fMRI/EEG comparison, these data are compared quantitatively in the text and in Figure 5.

Regarding the suggestion to perform standard ’source localization’ analysis, see responses to Reviewer 1.

Section 3.2 Validation with fMRIFigure 4 While there is consistency across 5 subjects, there are also subtle and not-so-subtle differences.What to make out of them?

Discrepancies in activations patterns between individuals is a complex neuroscience question that we feel is well beyond the scope of this paper.

Section 3.2 Validation with fMRIFigures 5 & 6 Figure 5 is also a qualitative figure from two subjects with no appropriate quantification of results across subjects. The same is true for Figure 6.

On the contrary, Figure 5 contains a quantitative comparison, which is now also described in the text. A quantitative comparison for the epilepsy data in Fig 6 (and C.4-C.6) is now shown in Fig 7.

Section 3.2 Validation with fMRIGiven the absence of appropriate “validation” of the proposed model and method, it is unclear how much one can trust results in Section 4.

We believe that the quantitative comparisons extant in the original text (and apparently missed by the Reviewer) along with the additional quantitative comparisons are sufficient to merit trust in Section 4.

Section 3.2 Validation with fMRIWhat are the thresholds used in maps for Figure 7? Was correction for multiple comparisons performed? The final arguments at the end of section 4 do not make sense. Is the claim that all results of reconstructions from SPECTRE shown here are significant with no reason for multiple comparison corrections to control for false positives? Why so?

We agree that the last line in Section 4 is misleading and have removed it.

Section 3.2 Validation with fMRIDiscussion is woefully inadequate in addition to the inconclusive findings presented here.

We have added a significant amount of text to the Discussion to address the points brought up by the Reviewer. And, contrary to the comments of this Reviewer, we believe the statistically significant results presented are not “inconclusive”.

Supplementary MaterialsThis reviewer had an incredibly difficult time understanding the inverse model solution. Even though this has been described in a prior publication by the authors, it is important and imperative that all details be provided here to make the current manuscript complete. The notation itself is so nonstandard. What is Σ^ij^, δ^ij^? Where is the reference for equation (1)? What about the equation for ^ˆ^(R)? There are very few details provided on the exact implementation details for the Fourier-space pseudo-spectral approach. What are the dimensions of the problem involved? How were different tissue compartments etc. handled? Equation 1 holds for the entire volume but the measurements are only made on the surface. How was this handled? What is the WETCOW brain wave model? I don’t see any entropy term defined anywhere - where is it?

We have added more detail on the theoretical and numerical aspects of the inverse problem in two new subsections “Theory” and “Numerical Implementation” in the new section “Solution to the inverse EEG problem”.

Supplementary MaterialsSo, how can one understand even at a high conceptual level what is being done with SPECTRE?

We have added a new subsection “Summary of SPECTRE” that provides a high conceptual level overview of the SPECTRE method outlined in the preceding sections.

Supplementary MaterialsIn order to understand what was being presented here, it required the reader to go on a tour of the many publications by the authors where the difficulty in understanding what they actually did in terms of inverse modeling remains highly obscure and presents a huge problem for replicability or reproducibility of the current work.

We have now included more basic material from our previous papers, and simplified the presentation to be more accessible. In particular, we have now moved the key aspects of the theoretic and numerical methods, in a more readable form, from the Supplementary Material to the main text, and added a new Appendix that provides a more intuitive and accessible overview of our estimation procedures.

Supplementary MaterialsHow were conductivity values for different tissue types assigned? Is there an assumption that the conductivity tensor is the same as the diffusion tensor? What does it mean that “in the present study only HRA data were used in the estimation procedure?” Does that mean that diffusion MRI data was not used? What is SYMREG? If this refers to the MRM paper from the authors in 2018, that paper does not include EEG data at all. So, things are unclear here.

The conductivity tensor is not exactly the same as the diffusion tensor in brain tissues, but they are closely related. While both tensors describe transport properties in brain tissue, they represent different physical processes. The conductivity tensor is often assumed to share the same eigenvectors as the diffusion tensor. There is a strong linear relationship between the conductivity and diffusion tensor eigenvalues, as supported by theoretical models and experimental measurements. For the current study we only used the anatomical data for estimatition and assignment of different tissue types and no diffusion MRI data was used. To register between different modalities, including MNI, HRA, function MRI, etc., and to transform the tissue assignment into an appropriate space we used the SYMREG registration method. A comment to the effect has been added to the text.

Supplementary MaterialsHow can reconstructed volumetric time-series of potential be thought of as the EM equivalent of an fMRI dataset? This sentence doesn’t make sense.

This sentence indeed did not make sense and has been removed.

Supplementary MaterialsTypical Bayesian inference does not include entropy terms, and entropy estimation doesn’t always lend to computing full posterior distributions. What is an “entropy spectrum pathway”? What is µ∗? Why can’t things be made clear to the reader, instead of incredible jargon used here? How does section 6.1.2 relate back to the previous section?

That is correct that Bayesian inference typically does not include entropy terms. We believe that their introduction via the theory of entropy spectrum pathways (ESP) is a significant advance in Bayesian estimation as it provides highly relevent prior information from within the data itself (and therefore always available in spatiotemporal data) that facilitates a practical methodology for the analysis of complex non-linear dynamical system, as contained in the entropy field decomposition (EFD).

Section 6.1.3 has now been replaced by a new Appendix A that discusses ESP in a much more intuitive and conceptual manner.

Supplementary MaterialsSection 6.1.3 describes entropy field decomposition in very general terms. What is “non-period”? This section is incomprehensible. Without reference to exactly where in the process this procedure is deployed it is extremely difficult to follow. There seems to be an abuse of notation of using ϕ for eigenvectors in equation (5) and potentials earlier. How do equations 9-11 relate back to the original problem being solved in section 6.1.1? What are multiple modalities being described here that require JESTER?

Section 6.1.3 has now been replaced by a new Appendix A that covers this material in a much more intuitive and conceptual manner.

Supplementary MaterialsSection 6.3 discusses source localization methods. While most forward lead-field models assume quasistatic approximations to Maxwell’s equations, these are perfectly valid for the frequency content of brain activity being measured with EEG or MEG. Even with quasi-static lead fields, the solutions can have frequency dependence due to the data having frequency dependence. Solutions do not have to be insensitive to detailed spatially variable electrical properties of the tissues. For instance, if a FEM model was used to compute the forward model, this model will indeed be sensitive to the spatially variable and anisotropic electrical properties. This issue is not even acknowledged.

The frequency dependence of the tissue properties is not the issue. Our theoretical work demonstrates that taking into account the anisotropy and inhomogeneity of the tissue is necessary in order to derive the existence of the weakly evanescent transverse cortical waves (WETCOW) that SPECTRE is detecting. We have added more details about the WETCOW model in the new Section “A physical theory of brain wave” to emphasize this point.

Supplementary MaterialsArguments to disambiguate deep vs shallow sources can be achieved with some but not all source localization algorithms and do not require a non-quasi-static formulation. LORETA is not even the main standard algorithm for comparison. It is disappointing that there are no comparisons to source localization and that this is dismissed away due to some coding issues.

Again, we are not doing ’source localization’. The concept of localized dipole sources is anathema to our brain wave model, and so in our view comparing SPECTRE to such methods only propagates the misleading idea that they are doing the same thing. So they are definitely not dismissed due to coding issues. However, because of repeated requests to do compare SPECTRE with such methods, we attempted to run a standard source localization method with parameters that would at least provide the closest approximation to what we were doing. This attempt highlighted a serious computational issue in source localization methods that is a direct consequence of the fact that they are not attempting to do what SPECTRE is doing - describing a time-varying wave field, in the technical definition of a ’field’ as an object that has a value at every point in space-time.